# Generalisation under gradient descent via deterministic PAC-Bayes

**Eugenio Clerico**[*][†]                                   EUGENIO.CLERICO@GMAIL.COM
*Universitat Pompeu Fabra, Barcelona*

**Tyler Farghly**[*]                                          FARGHLY@STATS.OX.AC.UK
*Department of Statistics, University of Oxford*

**George Deligiannidis**                          DELIGIAN@STATS.OX.AC.UK
*Department of Statistics, University of Oxford*

**Benjamin Guedj**                                         B.GUEDJ@UCL.AC.UK
*AI Centre and Department of Computer Science, UCL London & Inria London*

**Arnaud Doucet**                                       DOUCET@STATS.OX.AC.UK
*Department of Statistics, University of Oxford*

**Editors:** Gautam Kamath and Po-Ling Loh

## Abstract

We establish disintegrated PAC-Bayesian generalisation bounds for models trained with gradient descent methods or continuous gradient flows. Contrary to standard practice in the PAC-Bayesian setting, our result applies to optimisation algorithms that are deterministic, without requiring any *de-randomisation* step. Our bounds are fully computable, depending on the density of the initial distribution and the Hessian of the training objective over the trajectory. We show that our framework can be applied to a variety of iterative optimisation algorithms, including stochastic gradient descent (SGD), momentum-based schemes, and damped Hamiltonian dynamics.

## 1. Introduction

Effectively upper bounding the generalisation error of modern learning algorithms is an open problem of great importance to the statistical learning theory community (Zhang et al., 2016). Originally, properties of the hypothesis space, such as VC dimension and Rademacher complexity (Vapnik, 2000; Bousquet et al., 2004; Shalev-Shwartz and Ben-David, 2014), were used to establish *worst-case* generalisation bounds, holding uniformly over all possible algorithms and training datasets. However, as these results are often vacuous in over-parameterised settings, the modern perspective focuses on algorithm and data-dependent bounds (McAllester, 1998; Bousquet and Elisseeff, 2002; Hardt et al., 2016; Xu and Raginsky, 2017; Clerico et al., 2022b; Lugosi and Neu, 2022).

Among the various approaches, the PAC-Bayesian framework (Guedj, 2019; Alquier, 2021) has obtained particularly promising empirical results (Dziugaite and Roy, 2017; Zhou et al., 2019; Pérez-Ortiz et al., 2021a,b; Biggs and Guedj, 2022b; Clerico et al., 2022a). Typically, a PAC-Bayes bound is an upper bound on the expected population loss of a stochastic algorithm, holding with high probability on the random draw of the training dataset. This framework gained popularity after yielding non-vacuous empirical bounds in overparameterised regimes, such as modern neural networks (Dziugaite and Roy, 2017; Zhou et al., 2019; Pérez-Ortiz et al., 2021a; Clerico et al., 2022a). Since the standard PAC-Bayesian framework relies on the randomness of the trainable

---

[*] The first two authors contributed equally.

[†] This work was carried out while the author was affiliated with the Department of Statistics, University of Oxford.

parameters, this type of analysis is typically applied to specifically designed stochastic models. For instance, in the setting of neural networks, this requires an architecture featuring stochastic weights and biases, instead of the standard deterministic ones. To extend these ideas to deterministic settings, de-randomisation techniques are used. One possibility is to leverage stability properties to approximate a model by randomly perturbing its parameters. While this approach has shown promising results for feed-forward neural networks (Neyshabur et al., 2018; Nagarajan and Kolter, 2019; Miyaguchi, 2019; Banerjee et al., 2020), it relies on specific architectural assumptions. An alternative way to tackle the problem provides bounds for the predictor obtained by averaging a stochastic one, an approach started by Germain et al. (2009). However, this leads to results that only apply to models with very specific structures: for instance, Letarte et al. (2019), Biggs and Guedj (2021), and Biggs and Guedj (2022a) obtained bounds for particular deterministic networks with a rather unusual erf activation function. Finally, besides PAC-Bayes bounds in expectation, there are disintegrated results that hold with high probability on a random realisation of the stochastic model (Catoni, 2004, 2007; Blanchard and Fleuret, 2007; Alquier and Biau, 2013; Guedj and Alquier, 2013; Rivasplata et al., 2020; Viallard et al., 2021). To the best of our knowledge, this last approach has not been applied to standard non-stochastic algorithms, such as neural networks trained via gradient descent methods.

Here, we consider models trained by gradient descent-type methods and leverage the framework of disintegrated PAC-Bayes bounds. We start by noticing that often training with a deterministic optimisation scheme does still involve some randomness, due to an initialisation that features a random draw of the initial values of the parameter (*e.g.*, most neural networks (Goodfellow et al., 2016)). Our analysis shows that it is possible to exploit this source of noise and obtain disintegrated PAC-Bayes bounds, holding with high probability on the random training dataset and initialisation. To the best of our knowledge, this is the first PAC-Bayesian result that directly applies to standard non-stochastic settings, without strong requirements on the model or the need for any randomness other than the initialisation. Besides, unlike bounds based on de-randomisation, ours apply with only limited assumptions made about the smoothness of the training objective, and can be computed in closed form using information collected along the trajectory of the parameters during training.

We compare our bounds with other known results, including some outside the scope of the PAC-Bayesian literature. When compared to uniform stability bounds (Elisseeff, 2005; Hardt et al., 2016; Bousquet et al., 2020), we find that ours have sharper rates with respect to the size of the training dataset, and grow slower with the number of iterations. With regards to the recently popularised information-theoretic bounds (Xu and Raginsky, 2017; Negrea et al., 2019; Neu et al., 2021; Clerico et al., 2022b), ours are noticeably easier to compute and are not limited to bounds in expectation. Evaluating our bound only requires knowledge of the density of the initial distribution and of the Hessian of the training objective over the optimisation trajectory. The latter captures the flatness of the optimisation objective along the training path and can be seen to agree with the notion that flatter minima generalise better (Hochreiter and Schmidhuber, 1997; Keskar et al., 2017; Izmailov et al., 2018; He et al., 2019; Neu et al., 2021). We also highlight here how this term relates to the implicit regularisation occurring in algorithms known to result in improved generalisation (Blanc et al., 2020; Damian et al., 2021). We demonstrate that this framework is easily extended to almost all iterative schemes, including stochastic variants of gradient descent (Kiefer and Wolfowitz, 1952), and iterative procedures based on auxiliary variables, like momentum schemes (Qian, 1999) or damped Hamiltonian dynamics (Hairer et al., 2006; França et al., 2020).

## 2. Notation and setting

We consider the standard supervised learning framework, where examples are pairs instance-label $z = (x, y) \in \mathcal{X} \times \mathcal{Y} = \mathcal{Z}$. A learning algorithm takes a training dataset $s = \{z_1, \ldots, z_m\}$ of $m$ examples and outputs a function $f : \mathcal{X} \to \mathcal{Y}$. More specifically, we consider algorithms that choose a hypothesis $h \in \mathcal{H}$, which is understood to parameterise a map $f_h : \mathcal{X} \to \mathcal{Y}$ (*e.g.*, $h$ could be the weights of a neural network). We always assume that $\mathcal{H} \subseteq \mathbb{R}^d$, for some dimension $d > 0$. We call the algorithm stochastic when its output $h$ is a random variable on $\mathcal{H}$, whose law can depend on $s$.

Given a loss function $\ell : \mathcal{H} \times \mathcal{Z} \to \mathbb{R}$, we define the empirical loss on a dataset $s$ by

$$\mathcal{L}_s(h) = \frac{1}{m} \sum_{z \in s} \ell(h, z) \,.$$

However, what often matters is how well $h$ predicts the labels of instances outside of $s$. Assuming that the population of examples follows a distribution $\mu$, the relevant quantity is the population loss,

$$\mathcal{L}_\mathcal{Z}(h) = \int_\mathcal{Z} \ell(h, z) \mathrm{d}\mu(z) \,.$$

Upper bounding $\mathcal{L}_\mathcal{Z}$ only knowing $\mathcal{L}_s$ is the subject of focus in this paper and in the literature on generalisation bounds more broadly. We assume that $s \sim \mu^m = \mu^{\otimes m}$ (*i.e.*, i.i.d. draws from $\mu$). We are interested in upper bounds on $\mathcal{L}_\mathcal{Z}(h)$ (with $h$ the hypothesis picked by the algorithm) holding with high probability on the random draw of $s$ (or on $(h, s)$ in the stochastic setting).

Our results are inspired and naturally find their place within the PAC-Bayesian framework. Although the main focus in the PAC-Bayes literature has been on bounds in expectation, Rivasplata et al. (2020) and Viallard et al. (2021) have recently brought back interest in disintegrated bounds, which actually date back to Catoni (2004, 2007) and Blanchard and Fleuret (2007). We refer to Alquier (2021) for an introductory exposition on PAC-Bayes that also discusses a few disintegrated results. PAC-Bayes bounds deal with a stochastic model that, given $s \sim \mu^m$, returns a random hypothesis $h \sim \rho^s$, where the superscript $^s$ stresses $\rho$'s dependence on $s$.[1] We call $\rho^s$ the *posterior* distribution and denote the joint law of $(s, h)$ as $\mu^m * \rho^s$, *i.e.*, $\mathrm{d}(\mu^m * \rho^s)(s, h) = \mathrm{d}\mu^m(s)\mathrm{d}\rho^s(h)$. A disintegrated PAC-Bayes bound is an upper bound on $\mathcal{L}_\mathcal{Z}(h)$ that holds with high probability over $(s, h) \sim \mu^m * \rho^s$. A fundamental ingredient in this framework is the comparison of the posterior $\rho^s$ with a *prior* distribution $\pi$ on $\mathcal{H}$, only required to be chosen without knowledge of the training dataset $s$. We write $\mu^m \otimes \pi$ for the law of a pair $(s, h)$, where $s \sim \mu^m$ and $h \sim \pi$ are independent.

This work provides generalisation bounds for algorithms whose output is obtained optimising an objective $\mathcal{C}_s : \mathcal{H} \to \mathbb{R}$ via gradient-based descent methods. $\mathcal{C}_s$ can depend on the training dataset $s$ and, in practice, it can coincide with the empirical loss. However, this is not necessarily the case, as one might use a surrogate loss for the training or add some regularising term. In our analysis, we use $h_t$ (or $h_k$) to denote the parameters at time $t$ (or iteration $k$). Similarly, we use $\rho_t$ (or $\rho_k$) to denote their marginal distribution. All the measures that we consider are absolutely continuous with respect to the Lebesgue measure, and we use the same notation to denote their density. The random initialisation is given by $\rho_0$, that we assume to have strictly positive density on the whole $\mathcal{H}$.

---

1. To be rigorous, one should actually require that $s \mapsto \rho^s$ is a Markov kernel.

## 3. Disintegrated PAC-Bayes for continuous-time gradient flows

We begin by considering the continuous-time dynamics of the gradient flow. While this setting is less realistic than the one considered in the discrete-time analysis to follow, the discussion is considerably cleaner and will help expose some of the primary ideas of the framework we propose. We define the gradient flow $(h_0, t) \mapsto \Phi_t^s(h_0) \in \mathcal{H}$ as the solution to the differential equation

$$\partial_t \Phi_t^s(h_0) = -\nabla \mathcal{C}_s(\Phi_t^s(h_0)); \qquad \Phi_0^s(h_0) = h_0. \qquad (1)$$

We assume that $\mathcal{C}_s$ is such that a solution exists until a fixed time horizon $T > 0$, for all $h_0 \in \mathcal{H}$ and training datasets, $s \in \text{supp}(\mu)^{\otimes m}$. Since $h_0$ and $s$ are fixed prior to training, we will simplify the notation, using $h_t$ to denote the solution of (1). Given a random initialisation $\rho_0$ we define $\rho_t$ as the push-forward of $\rho_0$ under the gradient flow:

$$\rho_t = \Phi_t^{s\#} \rho_0.$$

By sampling the initial parameters $h_0$ from $\rho_0$ and following the flow dynamics up to $T$, we get a hypothesis $h_T$ that is distributed according to $\rho_T$.

We take a PAC-Bayesian approach to deriving generalisation bounds, selecting $\rho_0$ as the prior and $\rho_T$ as the posterior distribution. With this, we obtain PAC-Bayesian generalisation bounds for an algorithm that, once $s$ and $h_0$ are drawn, is deterministic.

**Theorem 1** *Consider the dynamics $\partial_t h_t = -\nabla \mathcal{C}_s(h_t)$, where $\mathcal{C}_s : \mathcal{H} \to \mathbb{R}$ is twice differentiable, and let $\Psi : \mathbb{R}^2 \to \mathbb{R}$ be an arbitrary measurable function. Taking $\delta \in (0, 1)$ and $T > 0$ fixed, with probability at least $1 - \delta$ on the random draw $(s, h_0) \sim \mu^m \otimes \rho_0$, it holds that*

$$\Psi\big(\mathcal{L}_s(h_T), \mathcal{L}_{\mathcal{Z}}(h_T)\big) \leq \frac{1}{m} \left( \log \frac{\rho_0(h_0)}{\rho_0(h_T)} + \int_0^T \Delta \mathcal{C}_s(h_t) dt + \log \frac{\xi}{\delta} \right), \qquad (2)$$

*where $\Delta$ denotes the Laplacian with respect to $h$ and $\xi = \int_{\mathcal{Z}^m \times \mathcal{H}} e^{m \Psi(\mathcal{L}_{\bar{s}}(h), \mathcal{L}_{\mathcal{Z}}(h))} d\mu^m(\bar{s}) d\rho_0(h)$.*

How one chooses $\Psi$ is dependent on the integrability of $e^{m\Psi(\mathcal{L}_{\bar{s}}, \mathcal{L}_{\mathcal{Z}})}$ and thus, is dependent on properties of the loss function, initial distribution and data distribution. In the following corollaries, we provide two concrete settings in which $\Psi$ can be chosen to obtain more explicit bounds. In Corollary 2, we consider the setting in which the loss function is sub-Gaussian in $h$, setting $\Psi(u, v) = (v - u)/\sqrt{m}$. When $\ell$ is bounded a tighter bound holds and we can obtain faster rates using the approach of Langford and Seeger (2001) and Maurer (2004), taking $\Psi$ to be $m$ times the relative entropy between two Bernoulli distributions $\text{kl}(u\|v) = u \log \frac{u}{v} + (1 - u) \log \frac{1-u}{1-v}$. We remark that these choices of $\Psi$ are common in the PAC-Bayesian literature (Bégin et al., 2016; Alquier, 2021).

**Corollary 2** *Assume that $\ell(h, \cdot)$ is $R$-sub-Gaussian[2] for each $h \in \mathcal{H}$. Then, for any $\delta \in (0, 1)$ and $T > 0$, with a probability of at least $1 - \delta$ on the random draw $(s, h_0) \sim \mu^m \otimes \rho_0$, we have*

$$\mathcal{L}_{\mathcal{Z}}(h_T) \leq \mathcal{L}_s(h_T) + \frac{1}{\sqrt{m}} \left( \log \frac{\rho_0(h_0)}{\rho_0(h_T)} + \int_0^T \Delta \mathcal{C}_s(h_t) dt + \log \frac{1}{\delta} + \frac{R^2}{2} \right).$$

---

2. $\ell(h, \cdot)$ is $R$-sub-Gaussian if for all $\lambda \in \mathbb{R}$ we have $\log \int_{\mathcal{Z}} e^{\lambda \ell(h,z)} d\mu(z) \leq \lambda \int_{\mathcal{Z}} \ell(h, z) d\mu(z) + \frac{R^2 \lambda^2}{2}$.

**Corollary 3** *Assume that $\ell$ is bounded in $[0, 1]$ and $m \geq 8$. Then, for any fixed $\delta \in (0, 1)$ and $T > 0$, with a probability of at least $1 - \delta$ on the random draw $(s, h_0) \sim \mu^m \otimes \rho_0$, we have*

$$\mathcal{L}_{\mathcal{Z}}(h_T) \leq \mathrm{kl}^{-1}\left(\mathcal{L}_s(h_T)\middle| \frac{B_T}{m}\right); \quad B_T = \log\frac{\rho_0(h_0)}{\rho_0(h_T)} + \int_0^T \Delta\mathcal{C}_s(h_t)\mathrm{dt} + \log\frac{2\sqrt{m}}{\delta}, \quad (3)$$

*where we define $\mathrm{kl}^{-1}(u|c) = \sup\{v \in [0, 1] : \mathrm{kl}(u\|v) \leq c\}$. In particular, it follows that*

$$\mathcal{L}_{\mathcal{Z}}(h_T) \leq \mathcal{L}_s(h_T) + \sqrt{\frac{\mathcal{L}_s(h_T)B_T}{2m}} + \frac{B_T}{2m}. \quad (4)$$

We note that as long as $B_T = O(\log m)$, one can expect standard rates of order $O(\sqrt{\log m/m})$. However, an advantage of the kl formulation is that it can yields faster rates if the empirical loss is controlled by $\mathcal{L}_s(h_T) = O(1/m)$ (*i.e.*, when the model is able to fit very well the training data). Indeed, in such case (4) is a fast-rate bound $O(\log m/m)$ (see, *e.g.*, the discussion after Theorem 1 in Tolstikhin and Seldin, 2013 or in Section 2 of Mhammedi et al., 2019). In Appendix B, we derive additional bounds for settings with weaker concentration guarantees, including sub-exponential concentration, by making use of recent results from Casado et al. (2024).

We remark that, in Theorem 1, the time horizon $T$ must be chosen prior to training and cannot depend on $s$ and $h_0$. However, the result can be generalised to allow for the algorithm to pick the best time horizon among a set of $\kappa$ candidates, $\{T_1, \ldots, T_\kappa\}$, only suffering an additional penalty of $\log \kappa$ on the right-hand side of (2) (*i.e.*, replacing $\log\frac{\xi}{\delta}$ with $\log\frac{\kappa\xi}{\delta}$). This follows from an elementary union argument where for each $T_k$, we consider a bound holding with probability at least $1 - \delta/\kappa$. As a final comment, we mention that the RHS of the bound in Theorem 1 diverges as $T \to \infty$, which reflects the fact that PAC-Bayes bounds are vacuous for degenerate posteriors.

**Proof of Theorem 1** The proof is based on two steps. First, we keep track of how the density evolves during training, obtaining an explicit expression for the posterior density. Then, we apply this in combination with a classical Markov's inequality argument.

For the first step, from the continuity equation[3] of the gradient flow, we have that

$$\partial_t \rho_t(h) = \nabla \cdot (\rho_t(h)\nabla\mathcal{C}_s(h)), \quad \text{for all } h \in \mathcal{H}, \quad (5)$$

and furthermore, we obtain that $\rho_t$ admits a Lebesgue density for all $t \in [0, T]$. From this, we obtain

$$\partial_t(\rho_t(h_t)) = \partial_t\rho_t(h_t) + \nabla\rho_t(h_t) \cdot \partial_t h_t = \rho_t(h_t)\Delta\mathcal{C}_s(h_t),$$

and in particular,

$$\log\frac{\rho_T(h_T)}{\rho_0(h_0)} = \int_0^T \Delta\mathcal{C}_s(h_t)\mathrm{dt}.$$

For the second step, it follows from Markov's inequality that with a probability of at least $1 - \delta$ on $(s, h_T) \sim \mu^m * \rho_T$, it holds that

$$e^{m\Psi(\mathcal{L}_s(h_T), \mathcal{L}_{\mathcal{Z}}(h_T)) - \log\frac{\rho_T(h_T)}{\rho_0(h_T)}} \leq \frac{1}{\delta}\int_{\mathcal{Z}^m \times \mathcal{H}} e^{m\Psi(\mathcal{L}_{\bar{s}}(h), \mathcal{L}_{\mathcal{Z}}(h)) - \log\frac{\rho_T(h)}{\rho_0(h)}} \mathrm{d}\mu^m(\bar{s})\mathrm{d}\rho_T(h).$$

---

3. The *continuity equation* is a key tool in studying the local evolution of the density (see, *e.g.*, Chapter 8 in Ambrosio et al., 2008). It comes from fluid dynamics and expresses the fact that the local rate of change of the density of a fluid equals the negative divergence of the flux. In our case, the flux can be expressed as minus the gradient of the training objective, leading to (5). Note that (5) can also be seen as a simplified version of the classical Fokker-Planck equation, in the absence of the Brownian noise.

Combining this with the fact that, for all $\bar{s} \in \mathcal{Z}^m$,

$$\int_{\mathcal{H}} \frac{\rho_0(h)}{\rho_T(h)} e^{m\Psi(\mathcal{L}_{\bar{s}}(h), \mathcal{L}_{\mathcal{Z}}(h))} \mathrm{d}\rho_T(h) = \int_{\{\rho_T > 0\}} e^{m\Psi(\mathcal{L}_{\bar{s}}(h), \mathcal{L}_{\mathcal{Z}}(h))} \mathrm{d}\rho_0(h),$$

we obtain

$$m\Psi(\mathcal{L}_{\bar{s}}(h_T), \mathcal{L}_{\mathcal{Z}}(h_T)) \leq \log \frac{\rho_T(h_T)}{\rho_0(h_T)} + \log \frac{\xi}{\delta},$$

with a probability of at least $1 - \delta$ on $(s, h_T) \sim \mu^m * \rho_T$. Since sampling $(s, h_T)$ from $\mu^m * \rho_T$ is equivalent to drawing $(s, h_0) \sim \mu^m \otimes \rho_0$ and following the dynamics up to $T$, the bound equivalently holds with a probability of at least $1 - \delta$ on $(s, h_0) \sim \mu^m \otimes \rho_0$. Using that

$$\log \frac{\rho_T(h_T)}{\rho_0(h_T)} = \log \frac{\rho_0(h_0)}{\rho_0(h_T)} + \log \frac{\rho_T(h_T)}{\rho_0(h_0)} = \log \frac{\rho_0(h_0)}{\rho_0(h_T)} + \int_0^T \Delta\mathcal{C}_s(h_t)\mathrm{d}t$$

we obtain the bound in the statement. ∎

## 4. Discrete time dynamics

In this section, we consider the gradient descent (GD) algorithm,

$$h_{k+1} = h_k - \eta_k \nabla \mathcal{C}_s(h_k),$$

where $\{\eta_k\}_{k=0}^{K-1}$ is the training schedule and the number of iterations, $K \geq 1$, is fixed. We let $\rho_k$ be the law of $h_k$, and assume that $\rho_0$ admits a positive Lebesgue density on the whole $\mathcal{H}$.

The primary obstacle in reproducing the methodology of the previous section is that we can no longer use the gradient flow continuity equation to keep track of the density change along the trajectory. However, the change in density can still be computed exactly as long as we can ensure that the update map is injective and differentiable along the path.

**Theorem 4** *For any dataset $s \in \mathcal{Z}^m$, assume there is a Borel set $A_s \subseteq \mathcal{H}$ on which $\mathcal{C}_s$ is twice-differentiable and $M$-smooth, where $\sup_k \eta_k \leq 1/(2M)$. Let $\Psi : \mathbb{R}^2 \to \mathbb{R}$ be measurable. Taking $\delta \in (0, 1)$ and $K \in \mathbb{N}$ fixed, suppose that $\{h_k\}_{k=0}^{K-1}$ lies in $A_s$ with a probability of at least $1 - \delta/2$ under $(s, h_0) \sim \mu^m \otimes \rho_0$. Then, with a probability of at least $1 - \delta$ on $(s, h_0) \sim \mu^m \otimes \rho_0$, it holds that*

$$\Psi(\mathcal{L}_s(h_T), \mathcal{L}_{\mathcal{Z}}(h_T)) \leq \frac{1}{m}\left(\log \frac{\rho_0(h_0)}{\rho_0(h_k)} - \sum_{k=0}^{K-1} \mathrm{tr}\log\left(\mathrm{Id} - \eta_k \nabla^2 \mathcal{C}_s(h_k)\right) + \log \frac{2\xi}{\delta} + \delta\right), \quad (6)$$

*where $\xi = \int_{\mathcal{Z}^m \times \mathcal{H}} e^{m\Psi(\mathcal{L}_{\bar{s}}(h), \mathcal{L}_{\mathcal{Z}}(h))} \mathrm{d}\mu^m(\bar{s})\mathrm{d}\rho_0(h)$.*

We refer to Appendix A for the proof. We note that the term $-\mathrm{tr}\log(\mathrm{Id} - \eta_k \nabla^2 \mathcal{C}_s(h_k))$ in (6) can be upper bounded in various ways, using the fact that $\mathcal{C}_s$ is smooth around $h_k$.

**Lemma 5** *With the notation of Theorem 4, let $h_k \in A_s$. Then*

$$-\mathrm{tr}\log\left(\mathrm{Id} - \eta_k \nabla^2 \mathcal{C}_s(h_k)\right) \leq \eta_k \Delta\mathcal{C}_s(h_k) + \eta_k^2 \|\nabla^2 \mathcal{C}_s(h_k)\|_{\mathrm{F}}^2 \leq \frac{3}{2}\eta_k \|\nabla^2 \mathcal{C}_s(h_k)\|_{\mathrm{TR}},$$

*where $\|\cdot\|_{\mathrm{F}}$ is the Frobenius norm and $\|\cdot\|_{\mathrm{TR}}$ the trace norm.[4]*

---

4. For a matrix $U$ with singular values $\{\sigma_i\}$, let $\|U\|_{\mathrm{TR}} = \sum_i \sigma_i$ and $\|U\|_{\mathrm{F}} = \sqrt{\sum_i \sigma_i^2} = \mathrm{tr}[UU^\top]^{1/2}$.

From the first inequality, we see that the continuous time bound of Theorem 1 is recovered when one takes the learning rate to $0$. From the final inequality, it follows that this term scales at worst as $O(d)$, as the smoothness assumption ensures that $\eta_k \|\nabla^2 \mathcal{C}_s(h_k)\|_{\mathrm{TR}} \leq d/2$. However, it is likely that in many cases, this translates to an overly pessimistic estimate. For instance, we show in Remark 18, in Appendix C.1, that for a simple random feature model one can upper bound $\|\nabla^2 \mathcal{C}_s\|_{\mathrm{TR}}$ with a term that is of order $O(1)$ for large $d$.

## 5. Examples with closed-form results

To complement the results of the previous sections, we consider toy examples for which more explicit forms for the bound can be obtained.

### 5.1. Random feature model

To start, we investigate a simple feature model trained via continuous-time gradient descent on a simple regression task. We let $\mathcal{X} = S^{p-1}$ be the unit sphere in $\mathbb{R}^p$ and $\mathcal{Y} = [0, 1]$. The goal is to learn a target function $f : \mathcal{X} \to \mathcal{Y}$. We consider the class of mappings $F_h : \mathcal{X} \to \mathbb{R}$ defined as

$$F_h(x) = \frac{1}{\sqrt{d}} h \cdot \Phi(x) ,\,^{[5]}$$

for $h \in \mathbb{R}^d$, where $\Phi : \mathbb{R}^p \to \mathbb{R}^d$ is a feature map, kept fixed during training. Since we know that $f$ takes values in $[0, 1]$, we let the prediction by the model be $f_h = \max\{0, \min\{1, F_h\}\}$. We consider a quadratic loss $\ell(h, z) = (f_h(x) - y)^2$, and an optimisation objective $\mathcal{C}_s(h) = \frac{1}{m} \sum_{z \in s} \hat{\ell}(F_h(x), y)$, with $\hat{\ell}(F, y) = (F - y)^2$. We focus the setting of random feature models (Rahimi and Recht, 2007; Mei and Montanari, 2022), where the features are given by $\Phi(x) = \phi(Wx)$. Here, $\phi$ is a non-linearity acting component-wise, and $W$ is a $d \times p$ matrix whose components are independently drawn from a standard Gaussian distribution. Interestingly, in the overparameterised regime $d \to \infty$ the bound does not become vacuous (see Appendix C.1 for the proof).

**Proposition 6** *For the random feature model described above, we have the limit in probability*

$$\lim_{d \to \infty} \left( \log \frac{\rho_0(h_0)}{\rho_0(h_T)} + \int_0^T \Delta \mathcal{C}_s(h_t) \mathrm{d}t \right) \leq T \left( \mathbb{E}_{\zeta \sim \mathcal{N}(0,1)}[\phi(\zeta)^2] + 2\sqrt{\mathcal{C}_s(h_0)} \right) .$$

The next natural question is how the bound scales with the dataset size $m$. In general, one cannot say much. Indeed, if $f$ is just random noise, it is clear that it is not possible to get a small population loss, although for long enough training time $T$ one could potentially learn the training dataset. However, we conjecture that if the target $f$ is *reasonably nice*, one can get a bound of order $\mathcal{L}_{\mathcal{Z}}(h_T) \lesssim \frac{\log m}{m}$ by selecting $T \sim \log m$. To be more explicit, we consider a simple setting, where $f$ is a linear combination of finitely many spherical harmonics (see Appendix C.1 for more details).

**Proposition 7** *Let $f : S^{d-1} \to [0, 1]$ be a linear combination of spherical harmonics of degree at most D. Let $\phi$ be given by $\phi(x) = \sum_{k=1}^J (x/e)^j$, with $J \geq D$.$^{[6]}$ Then, there exists a constant $\lambda > 0$,*

---

5. The factor $1/\sqrt{d}$ is a standard choice for this kind of models, as it brings $\|h/\sqrt{d}\| \sim O(1)$ at initialisation, when the components of $h$ are independently initialised $\mathcal{N}(0, 1)$ (*e.g.*, see Ghorbani et al., 2021 for random feature models).
6. In general, up to a zero-measure set of coefficients, any polynomial activation of degree $J$ would lead to the same result. The specific one picked here is to give an explicit example.

such that for $T = \frac{\log m}{4\lambda}$,

$$\lim_{d \to \infty} \mathcal{L}_{\mathcal{Z}}(h_T) \leq O_p\left(\frac{\log m}{m}\right), \quad \text{as } m \to \infty.$$

## 5.2. Wide neural networks

Next, we investigate the first term of the bound, $\log \rho_0(h_0)/\rho_0(h_k)$, by considering the setting of wide feed-forward neural networks. We assume that each layer has the same activation and the hidden layers have the same width $n$. Each of the weights and bias parameters are initialised with a centred Gaussian distribution with variance $\sigma_w^2/n$ and $\sigma_b^2$, respectively. We consider the quadratic loss as the optimisation objective and have the training schedule scale with $n$ such that $\eta_k = \bar{\eta}_k/n$, where $\bar{\eta}_k$ is independent of $n$. Thus, as $n$ grows, the evolution of the network can be approximated by linear dynamics according to the natural tangent kernel (NTK) (Jacot et al., 2018).

Borrowing the analysis of Lee et al. (2020), we obtain two properties of the large $n$ setting: (i) with high probability, $h_0$ (the initial value of all weights and biases) is such that $\mathcal{C}_s$ is smooth and bounded in a region around it and (ii) with the same probability, gradient descent stays close to initialisation. With this, we can show that this setting satisfies the assumptions of Theorem 4, and thus, we obtain the generalisation bound in the proposition that follows. We refer to Appendix C.2 for the formal statement and proof, as well as further discussion.

**Proposition 8 (Informal statement)** *Under suitable regularity conditions, for any $\delta \in (0,1)$, there exists $n_{\min}^\delta \in \mathbb{N}$ such that whenever $n \geq n_{\min}^\delta$, the assumptions of Theorem 4 are satisfied. In particular, with a probability of at least $1 - \delta$ on $(s, h_0) \sim \mu^m \otimes \rho_0$,*

$$\Psi\big(\mathcal{L}_s(h_K), \mathcal{L}_{\mathcal{Z}}(h_K)\big) \leq \frac{1}{m}\left(\frac{C}{\sigma_\omega^2} A_K(h_0) - \sum_{k=0}^{K-1} \operatorname{tr} \log\big(\operatorname{Id} - \eta_k \nabla^2 \mathcal{C}_s(h_k)\big) + \log \frac{2\xi}{\delta} + \delta\right),$$

*where we define,*

$$A_K(h_0) = \left(\left(\sum_{k=0}^{K-1} \bar{\eta}_k\right) \wedge \lambda_{\min}^{-1}\right)(J_s(h_0) + 1), \quad J_s(h_0)^2 = \frac{1}{m}\sum_{i=1}^m \langle h_0, \nabla_h F_{h_0}(x_i)\rangle,$$

*$\lambda_{\min}$ is the minimum eigenvalue of the NTK of the finite width network, $\xi$ is defined as in Theorem 4 and $C > 0$ is a constant.*

To better understand the nature of this bound, we discuss how both $A_K$, as well as the curvature dependent term could be further controlled in the limit as $n \to \infty$. In Remark 22, we show that a naive estimate can be used to obtain $J_s(h_0) = O_p(\sqrt{n})$ as $n \to \infty$. Furthermore, we argue that a central limit heuristic could be used to show that as $n \to \infty$, $J_s(h_0) = O_p(1)$. However, stating a formal argument in this direction is beyond the scope of this work. The Hessian of the NTK has been analysed in various recent works with a notable contribution from Jacot et al. (2020), who show that the Laplacian of the NTK training objective converges in the limit as $n \to \infty$ to the trace of the NTK and a term that decays over iterations. In Remark 23, we show that their arguments can also be applied to control the Hessian dependent term as $n \to \infty$. A consequence of this analysis is that for large $k$, with high probability, it holds that

$$\lim_{n \to \infty} -\operatorname{tr} \log\big(\operatorname{Id} - \eta_k \nabla^2 \mathcal{C}_s(h_k)\big) \leq 3\bar{\eta}_k \operatorname{tr}(\Theta)/m.$$

The bound given in Proposition 8 scales linearly with the number of iterations, and therefore, the rate of convergence of the training loss. This rate is governed by the spectrum of the NTK and therefore, it depends on the dataset. n the worst case, the convergence rate is controlled by the minimum eigenvalue of the NTK, making it is sufficient to have $\sum_{k=0}^{K-1} \bar{\eta}_k = \Omega(\lambda_{\min}^{-1} \log m)$. The spectrum of the NTK and more specifically, its minimum eigenvalue, has appeared in a variety of analyses of the NTK that have shown its relationship to the generalisation capabilities of the model (Arora et al., 2019) as well as memorisation properties (Montanari and Zhong, 2022).

## 6. Extension to other algorithms

While we have focused on gradient descent, the analysis can be extended to any iterative scheme

$$h_{k+1} = h_k + V_s(h_k; k),$$

where $V_s : \mathcal{H} \times \mathbb{N} \to \mathcal{H}$ can be any iteration-dependent vector field. This leads to the following generalisation of Theorem 4, which differs only in the trajectory dependent sum, where $V_s$ now replaces $-\nabla^2 \mathcal{C}_s$. A similar result is found for continuous flows (Theorem 24 in Appendix D).

**Theorem 9** *Consider the dynamics $h_{k+1} = h_k + V_s(h_k; k)$. For each dataset $s \in \mathcal{Z}^m$, denote as $A_s$ a Borel set on which $V_s$ is differentiable and $M$-Lipschitz, with $M \leq 1/2$. Let $\Psi : \mathbb{R}^2 \to \mathbb{R}$ be a measurable function. Fix $K \in \mathbb{N}$ and choose $\delta \in (0, 1)$, such that the trajectory $\{h_k\}_{k=0}^{K-1}$ lies in $A_s$ with probability at least $1 - \delta/2$, under $(s, h_0) \sim \mu^m \otimes \rho_0$. Then, with a probability of at least $1 - \delta$ on $(s, h_0) \sim \mu^m \otimes \rho_0$,*

$$\Psi\big(\mathcal{L}_s(h_K), \mathcal{L}_\mathcal{Z}(h_K)\big) \leq \frac{1}{m} \left( \log \frac{\rho_0(h_0)}{\rho_0(h_K)} - \sum_{k=0}^{K-1} \operatorname{tr} \log \Big( \operatorname{Id} + \nabla V_s(h_k; k) \Big) + \log \frac{2\xi}{\delta} + \delta \right),$$

*where $\nabla V_s$ is the Jacobian of $V_s$ with respect to $h$, and $\xi = \int_{\mathcal{Z}^m \times \mathcal{H}} e^{m\Psi(\mathcal{L}_{\bar{s}}(h), \mathcal{L}_\mathcal{Z}(h))} \mathrm{d}\mu^m(\bar{s}) \mathrm{d}\rho_0(h)$.*

**Stochastic gradient descent**  An immediate corollary of the above is that our theory applies to noisy variants of gradient descent with little modification. For example, we can consider a version of gradient descent that only evaluates $\mathcal{C}$ on a mini-batch $s_k \subset s$ at each iteration $k \in \mathbb{N}$, by simply setting $V_s(h; k) = -\eta_k \nabla \mathcal{C}_{s_k}(h)$. The resulting generalisation bound applies identically for stochastic variants of this scheme, including stochastic gradient descent, where the resulting bound is a function of the instance of the sampled mini-batches. To make things more explicit, we consider a surrogate loss function $\hat{\ell} : \mathcal{Z} \times \mathcal{H} \to \mathbb{R}$ and for a batch $s_k \subset s$, we write $\mathcal{C}_{s_k}(h) = \frac{1}{|s_k|} \sum_{z \in s_k} \hat{\ell}(z, h)$. For a sequence of batches $\{s_k\}$ (potentially randomly selected), we consider the dynamics $h_{k+1} = h_k - \eta_k \nabla \mathcal{C}_{s_k}(h_k)$. Then, under suitable smoothness assumptions for $\hat{\ell}$, we can derive from Theorem 9 the following bound

$$\Psi\big(\mathcal{L}_s(h_K), \mathcal{L}_\mathcal{Z}(h_K)\big) \leq \frac{1}{m} \left( \log \frac{\rho_0(h_0)}{\rho_0(h_K)} - \sum_{k=0}^{K-1} \operatorname{tr} \log \Big( \operatorname{Id} + \eta_k \Delta \mathcal{C}_{s_k}(h_k) \Big) + \log \frac{2\xi}{\delta} + \delta \right),$$

which holds with probability at least than $1 - \delta$ on the randomness of the training dataset $s$, the initialisation $h_0$, and the choice of the batches (see Proposition 25 in Appendix D.1).

**Momentum dynamics** We can also use Theorem 9 to consider settings in which auxiliary variables are used to compute the update. We do this by replacing $h_k$ with the pair $(h_k, v_k)$, where $v_k$ denotes the auxiliary variable. Indeed, this setting applies to a wide range of optimisation schemes. Note that in this scenario, the initial density $\rho_0$ must refer to the pair $(h_0, v_0)$.

An example is the momentum dynamics $h_{k+1} = h_k + v_{k+1}$ and $v_{k+1} = \mu_k v_k - \eta_k \nabla \mathcal{C}_s(h_k)$, for some momentum schedule $\{\mu_k\}$. In such a case we obtain a high probability bound in the form

$$\Psi\big(\mathcal{L}_s(h_K), \mathcal{L}_{\mathcal{Z}}(h_K)\big) \leq \frac{1}{m} \left( \log \frac{\rho_0(h_0, v_0)}{\rho_0(h_K, v_K)} - d \sum_{k=0}^{K-1} \log \frac{1}{\mu_k} + \log \frac{2\xi}{\delta} + \delta \right).$$

We refer to Appendix D.2 for further details and discussion.

**Damped Hamiltonian dynamics** For damped Hamiltonian dynamics (França et al., 2020), we can exploit the fact that the joint density of the 'position-momentum' pair $(h, v)$ is conserved under the Hamiltonian flow, a property preserved for discrete time-steps via symplectic integrators (Hairer et al., 2006). Using this, we obtain bounds for discrete-time algorithms without any smoothness assumptions on $\mathcal{C}_s$, other than twice differentiability. We refer to Appendix E for the details.

## 7. Comparison with the literature

### 7.1. Comparison with other PAC-Bayes bounds

Contrary to many PAC-Bayesian results, our bounds have the remarkable feature of applying to neural networks with deterministic parameters, trained via standard gradient-descent methods. Yet, this is not a complete novelty. A few previous works in the literature propose to study the generalisation of a deterministic network via the PAC-Bayes analysis of a noisy stochastic perturbation of it. This idea was exploited by Neyshabur et al. (2018), and later Biggs and Guedj (2022a), for a $L$-layer fully connected architecture with 1-Lipschitz homogeneous activation function and 1-Lipschitz loss. Combining margin arguments with PAC-Bayes techniques they found (up to logarithmic factors)

$$\mathcal{L}_{\mathcal{Z}}(h) \leq \mathcal{L}_s^{\gamma}(h) + O\big(\sqrt{dn\Gamma/(\gamma^2 m)}\big), \tag{7}$$

where $n$ is the width of the network, $\mathcal{L}_s^{\gamma}$ the margin empirical loss with margin $\gamma > 0$,[7] and $\Gamma = \sum_{l=1}^{L} \left( \|W_l\|_F^2 \prod_{l' \neq l} \|W_{l'}\| \right)$, with $\{W_l\}_{l=1}^{L}$ the weights of the network, $\|\cdot\|$ denoting the spectral norm, and $\|\cdot\|_F$ the Frobenius norm. One of the main issues of this result is $\Gamma$'s exponential dependence on the depth, due to the product of the norm of the weights. On the other hand, our bounds involve the Hessian term, which are at most of order $d$ (see Lemma 5), and the contribution $\log \frac{\rho_0(h_0)}{\rho_0(h_T)}$. When the weights are independently initialised as $\mathcal{N}(0, 1)$, this last term is upper bounded by $\frac{1}{2} \sum_{l=1}^{L} \|W_l\|_F^2$. Moreover, as shown in (4), for small enough empirical loss $\mathcal{L}_s = O(1/m)$, our bound can have a fast-rate dependence of $O(1/m)$ on the training dataset size.

Later, building on the ideas from Neyshabur et al. (2018), Nagarajan and Kolter (2019) obtained a bound that does not suffer of the exponential dependence on the depth. However, this comes at the price of inversely scaling with the smallest absolute value of the pre-activations on the training data,

---

7. The margin $\gamma$ is a standard measure of the confidence of the network's prediction. We refer to Neyshabur et al. (2018) or Biggs and Guedj (2021) for a definition of the margin loss. Note however that we always have $\mathcal{L}_s^{\gamma} \geq \mathcal{L}_s$.

leading to vacuous bounds in practice. We also mention that a result similar to that of Neyshabur et al. (2018) was previously established by Bartlett et al. (2017), without PAC-Bayes techniques.

The bounds discussed so far only take into account the final output of the algorithm, while our result looks at the evolution of the model during the training. A similar spirit is shared by Miyaguchi (2019), where the author focuses on the continuous time setting and studies the evolution of the generalisation gap under the gradient flow training dynamics. Applying this result to a multilayer network, it is possible to re-derive (7) under slightly weaker assumptions.

Most other PAC-Bayesian results for deterministic models cannot be applied to standard training algorithms, as they require strong, and often unusual, assumptions on the model architecture (*e.g.*, Letarte et al., 2019; Germain et al., 2009; Biggs and Guedj, 2021), or sampling the parameters from the posterior distribution (*e.g.*, Zantedeschi et al., 2021; Viallard et al., 2021; Rivasplata et al., 2020).

Finally, it is worth mentioning the work of Luo et al. (2022), which also studies de-randomised PAC-Bayesian guarantees for GD methods. However, it is hard to directly compare our findings with their results, as their framework and ours differ significantly. Specifically, their analysis focuses on discretised versions of GD and SGDm and their bounds are stated in terms of the gradient discretisation error, and become vacuous as this error approaches zero (standard GD/SGD). Moreover, their bounds require the prior distribution to depend on a subset of the training dataset. Overall, we believe that our results address more conventional versions of GD and SGD, albeit with the trade-off of requiring a smoothness assumption not needed in their work.

## 7.2. Comparison with the stability literature

Another method for obtaining algorithm-dependent generalisation bounds is the framework of uniform stability (Hardt et al., 2016; Pensia et al., 2018; Farghly and Rebeschini, 2021; Raj et al., 2023), proposed by Elisseeff (2005). This approach has recently received attention due to its application to fundamental optimisation methods, such as gradient descent and its stochastic counterpart (Hardt et al., 2016), while other works have leveraged it to obtain bounds in high-probability (Feldman and Vondrak, 2018, 2019; Bousquet et al., 2020). Hardt et al. (2016) considered the SGD training with non-convex training objectives. For $\mathcal{C}_s$ is $M$-smooth in $h$ (uniformly on $\mathcal{Z}$), $\ell$ is $L$-Lipschitz and bounded in $[0, 1]$, and the step-size satisfying $\eta_k \leq c/(k+1)$, the analysis of Hardt et al. (2016) and Bousquet et al. (2020) leads to the bound (with probability at least $1 - \delta$ and for some constant $C$)

$$\mathcal{L}_{\mathcal{Z}}(h) \leq \mathcal{L}_s(h) + C \left( \frac{(K/c)^{\frac{Mc}{Mc+1}} \log{(m/\delta)}}{Mm} + \sqrt{\frac{\log \delta^{-1}}{m}} \right), \tag{8}$$

To compare with these results, under the same assumptions, for GD Theorem 4 yields (cf. also (4))

$$\mathcal{L}_{\mathcal{Z}}(h_K) \leq \mathcal{L}_s(h_K) + \sqrt{\frac{\mathcal{L}_s(h_K)B_K}{2m}} + \frac{B_K}{2m},$$

with

$$B_K = \log \frac{\rho_0(h_0)}{\rho_0(h_K)} + \sum_{k=0}^{K-1} (\eta_k \Delta \mathcal{C}_s(h_k) + \eta_k^2 \|\nabla^2 \mathcal{C}_s(h_k)\|_{\mathrm{F}}^2) + \log{(m/\delta)}.\text{[8]}$$

Using Proposition 25 a similar bound can be obtained for SGD.

---

8. Note that since here we have that the smoothness holds on the whose $\mathcal{H}$, we do not need the additional term we can replace $\log(2\delta/m) + \delta$ in (6) with $\log(\delta/m)$.

As a first point of comparison, we note that our analysis does not require the Lipschitz assumption but only smoothness along the path of GD with high probability. Additionally, our analysis holds in both the stochastic and non-stochastic setting, whereas the technique used by Hardt et al. (2016) fundamentally requires random mini-batches to have bounds that decay with $m$. While the bound of (8) decays at a rate of $m^{-1/2}$, our bound can decays faster with rates $\log(m)/m$ (for $\mathcal{L}_s \sim 1/m$). The fact that $\sum_{k=0}^{K-1} \eta_k \sim c \log K$ suggests that our bound may scale better with $K$, though this would require the $\Delta\mathcal{C}_s(h_k)$ and $\|\nabla^2\mathcal{C}_s(h_k)\|_F$ terms to not grow too quickly with $k$. In the worst case, the smoothness can be used to upper bound these terms by $3dM/2$.

Lastly, we note that one of the main criticisms to the uniform stability approach is that it is solely related to the algorithm and does not consider specifics of the data or the distribution of the labels, raising doubts on its ability to distinguish whether a model has been trained on true or random labels (Zhang et al., 2016). On the other hand, our bound can depend on the data distribution through the optimisation objective $\mathcal{C}_s$ and its landscape along the training trajectory.

### 7.3. Comparison with information-theoretic bounds

Another popular direction within the literature on generalisation bounds uses ideas from information theory to upper bound the expected generalisation error in terms of the mutual information (Xu and Raginsky, 2017; Russo and Zou, 2019). This has been particularly practical for developing data-dependent bounds for noisy iterative methods, such as stochastic gradient Langevin dynamics and SGD (Mou et al., 2018; Negrea et al., 2019; Neu et al., 2021).

The general approach to this requires controlling the mutual information between the training data and the update of each iterate. Therefore, this technique is restricted to settings where noise is applied at each iteration, and the bounds explode when the amount of noise is reduced. To apply this to GD and SGD, Neu et al. (2021) consider a surrogate model trained by a Gaussian perturbation of these iterates. When the loss is $R$-sub-Gaussian, they bound the expected generalisation error by

$$\mathbb{E}\mathcal{L}_{\mathcal{Z}}(h_K) \leq \mathbb{E}\mathcal{L}_s(h_K) + \left(\frac{R^2}{m}\sum_{k=1}^{k}\eta_k^2\mathbb{E}A(h_k)\right)^{1/2} + |\mathbb{E}B(h_K)|,$$

where $A(h)$ and $B(h)$ measures the sensitivity of the gradient and the loss function, respectively, to perturbations in the parameters and dataset at $h$. A notable difference between this technique and our method is that this can only provide bounds in expectation. This comes with the downside that the right-hand side can usually not be computed exactly and the expectation of $A$ and $B$ should be approximated using a Monte Carlo average. In contrast, our bound is based on the instance of the optimisation trajectory, it can be computed exactly. However, our bounds have similar dependence on $m$ but worse dependence on $\eta_k$.

## 8. Conclusion

We derive novel high-probability generalisation bounds for models learned via optimisation algorithms such as gradient descent. Contrary to the standard PAC-Bayesian framework, our guarantees apply to models whose only randomness lies in the initialisation without requiring any *de-randomisation* step. To the best of our knowledge, our results are the first to leverage the disintegrated PAC-Bayesian framework to analyse such settings. We make this explicit by stating a bound that holds for wide neural networks trained via gradient descent.

A strength of our bounds is that it assumes little about the model or training procedure. For the continuous gradient flow dynamics, we require only that the optimisation objective be twice differentiable. For the discrete-time algorithm, we require smoothness and twice differentiability only in high probability on the trajectory. Additionally, we show that our results can be extended to settings more general than gradient descent and give explicit bounds for SGD, momentum schemes, and damped Hamiltonian dynamics. We foresee that this should motivate further work into developing generalisation bounds for other optimisation algorithms.

A promising direction for future work could be designing computationally efficient methods for computing these bounds. We would also like to evaluate the tightness of our guarantees and compare them with other results known in the literature with thorough empirical investigation. Finally, we believe that our results can be improved by identifying more easily verifiable assumptions to make the framework more broadly applicable.

## Acknowledgments

Eugenio Clerico was partially supported by the UK Engineering and Physical Sciences Research Council (EPSRC) through the grant EP/R513295/1 (DTP scheme) and Arnaud Doucet by EPSRC CoSInES EP/R034710/1. Tyler Farghly was supported by EPSRC EP/T5178 and by the DeepMind scholarship. Benjamin Guedj and Arnaud Doucet acknowledge support of the UK Defence Science and Technology Laboratory (DSTL) and EPSRC grant EP/R013616/1. This work was part of the collaboration between US DOD, UK MOD and UK EPSRC under the Multidisciplinary University Research Initiative. Benjamin Guedj acknowledges partial support from the French National Agency for Research, grants ANR-18-CE40-0016-01 and ANR-18-CE23-0015-02. The authors would like to thank Jake Fawkes, Shahine Bouabid, Umut Şimşekli, and Patrick Rebeschini for the valuable comments and suggestions.

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

## Appendix A. Omitted proofs of Sections 3 and 4

**Corollary 2** *Assume that $\ell(h, \cdot)$ is $R$-sub-Gaussian for each $h \in \mathcal{H}$. Then, for any $\delta \in (0, 1)$ and $T > 0$, with probability at least $1 - \delta$ on the random draw $(s, h_0) \sim \mu^m \otimes \rho_0$, we have*

$$\mathcal{L}_{\mathcal{Z}}(h_T) \leq \mathcal{L}_s(h_T) + \frac{1}{\sqrt{m}} \left( \log \frac{\rho_0(h_0)}{\rho_0(h_T)} + \int_0^T \Delta \mathcal{C}_s(h_t) \mathrm{d}t + \log \frac{1}{\delta} + \frac{R^2}{2} \right).$$

**Proof** The results follows from Theorem 1 with $\Psi(u, v) = (v - u)/\sqrt{m}$. In this case, we have that for each fixed $h$

$$\int_{\mathcal{X}} e^{m \Psi(\mathcal{L}_{\bar{s}}(h), \mathcal{L}_{\mathcal{Z}}(h))} \mathrm{d}\mu^m(\bar{s}) = \left( \int_{\mathcal{X}} e^{(\mathcal{L}_{\mathcal{Z}}(h) - \ell(h, z))/\sqrt{m}} \mathrm{d}\mu(z) \right)^m \leq e^{R^2/2},$$

by the definition of sub-Gaussianity. So, $\xi \leq R^2/2$, and the desired bound follows immediately. ∎

**Corollary 3** *Assume that $\ell$ is bounded in $[0, 1]$ and $m \geq 8$. Then, for any fixed $\delta \in (0, 1)$ and $T > 0$, with a probability of at least $1 - \delta$ on the random draw $(s, h_0) \sim \mu^m \otimes \rho_0$, we have*

$$\mathcal{L}_{\mathcal{Z}}(h_T) \leq \mathrm{kl}^{-1} \left( \mathcal{L}_s(h_T) \middle| \frac{B_T}{m} \right); \quad B_T = \log \frac{\rho_0(h_0)}{\rho_0(h_T)} + \int_0^T \Delta \mathcal{C}_s(h_t) \mathrm{d}t + \log \frac{2\sqrt{m}}{\delta},$$

*where we define $\mathrm{kl}^{-1}(u|c) = \sup\{v \in [0, 1] : \mathrm{kl}(u\|v) \leq c\}$. In particular, it follows that*

$$\mathcal{L}_{\mathcal{Z}}(h_T) \leq \mathcal{L}_s(h_T) + \sqrt{\frac{\mathcal{L}_s(h_T) B_T}{2m}} + \frac{B_T}{2m}.$$

**Proof** The bound follows from Theorem 1 with $\Psi(u, v) = \mathrm{kl}(u\|v)$, after using the fact that with this choice one has $\xi \leq 2\sqrt{m}$ if the loss is bounded in $[0, 1]$ and $m \geq 8$ (see, *e.g.*, Theorem 1 in Maurer, 2004). For the second bound, we use the property,

$$\mathrm{kl}^{-1}(u|c) \leq \min\{u + \sqrt{uc/2} + c/2, u + \sqrt{c/2}\},$$

(see, for example, Tolstikhin and Seldin 2013 and the discussion and references therein). ∎

**Theorem 4** *For any dataset $s \in \mathcal{Z}^m$, assume there is a Borel set $A_s \subseteq \mathcal{H}$ on which $\mathcal{C}_s$ is twice-differentiable and $M$-smooth, where $\sup_k \eta_k \leq 1/(2M)$. Let $\Psi : \mathbb{R}^2 \to \mathbb{R}$ be measurable. Taking $\delta \in (0, 1)$ and $K \in \mathbb{N}$ fixed, suppose that $\{h_k\}_{k=0}^{K-1}$ lies in $A_s$ with a probability of at least $1 - \delta/2$ under $(s, h_0) \sim \mu^m \otimes \rho_0$. Then, with a probability of at least $1 - \delta$ on $(s, h_0) \sim \mu^m \otimes \rho_0$, it holds that*

$$\Psi(\mathcal{L}_s(h_T), \mathcal{L}_{\mathcal{Z}}(h_T)) \leq \frac{1}{m} \left( \log \frac{\rho_0(h_0)}{\rho_0(h_k)} - \sum_{k=0}^{K-1} \mathrm{tr} \log \left( \mathrm{Id} - \eta_k \nabla^2 \mathcal{C}_s(h_k) \right) + \log \frac{2\xi}{\delta} + \delta \right),$$

*where $\xi = \int_{\mathcal{Z}^m \times \mathcal{H}} e^{m \Psi(\mathcal{L}_{\bar{s}}(h), \mathcal{L}_{\mathcal{Z}}(h))} \mathrm{d}\mu^m(\bar{s}) \mathrm{d}\rho_0(h)$.*

**Proof** First, for any $s \in \mathcal{Z}^m$, we introduce the notation $G_\eta^s$ for the mapping (from $\mathcal{H}$ to $\mathcal{H}$) $h \mapsto h - \eta \mathcal{C}_s(h)$. Then, define

$$A_s^0 = \{h_0 \in \mathcal{H} : h_k \in A_s \text{ for all } k \in \{0, \dots, K-1\}\},$$

which is a Borel set thanks to the regularity of the $G_{\eta_k}^s$ on $A_s$.

We start by noticing that for all $k$, the restriction of $G_{\eta_k}^s$ to $A_s$ is injective. Indeed, if $h$ and $h'$ are in $A_s$, we have that

$$\|G_{\eta_k}^s(h) - G_{\eta_k}^s(h')\| \ge \|h - h'\| - \eta_k \|\nabla \mathcal{C}_s(h) - \nabla \mathcal{C}_s(h')\| \ge \frac{1}{2}\|h - h'\|.$$

For any fixed $s$, if we condition on $h_0 \in A_s^0$, this condition is satisfied for all $k \in \{0, \dots, K-1\}$. Now let $\widetilde{\rho}_k^s = \rho_k(\cdot | h_0 \in A_s^0)$ be the law of $h_k$, conditioned on $h_0 \in A_s^0$. If we denote as $\widetilde{G}_{s,k}$ the restriction of $G_{\eta_k}^s$ to $\operatorname{supp} \widetilde{\rho}_k^s$, we have that $\widetilde{G}_{s,k}$ is a differentiable bijection $\operatorname{supp} \widetilde{\rho}_k^s \leftrightarrow \operatorname{supp} \widetilde{\rho}_{k+1}^s$. In particular, we see by induction that this implies that $\widetilde{\rho}_{k+1}^s$ admits a Lebesgue density (since $\widetilde{\rho}_0^s \ll \rho_0$), and by the change of variable formula

$$\widetilde{\rho}_{k+1}^s(h) = \widetilde{\rho}_k^s \circ \widetilde{G}_{s,k}^{-1}(h) \det\left[\operatorname{Id} - \eta_k \nabla^2 \mathcal{C}_s \circ \widetilde{G}_{s,k}^{-1}(h)\right]^{-1}, \quad \forall h \in \operatorname{supp} \widetilde{\rho}_{k+1}^s.$$

For $h_k \in \operatorname{supp}\widetilde{\rho}_k^s$, since $\widetilde{G}_{s,k}(h_k) = h_{k+1}$, we get

$$\widetilde{\rho}_{k+1}^s(h_{k+1}) = \widetilde{\rho}_k^s(h_k) \det\left[\operatorname{Id} - \eta_k \nabla^2 \mathcal{C}_s(h_k)\right]^{-1}.$$

In particular, for $h_0 \in A_s^0$ we have that $h_k \in \operatorname{supp}\widetilde{\rho}_k^s$ by definition of $\widetilde{\rho}_k^s$, and so

$$
\begin{aligned}
\log \frac{\widetilde{\rho}_K^s(h_K)}{\widetilde{\rho}_0^s(h_0)} &= \sum_{k=0}^{K-1} \log \frac{\widetilde{\rho}_{k+1}^s(h_{k+1})}{\widetilde{\rho}_k^s(h_k)} \\
&= -\sum_{k=0}^{K-1} \log \det\left[\operatorname{Id} - \eta_k \nabla^2 \mathcal{C}_s(h_k)\right] = -\sum_{k=0}^{K-1} \operatorname{tr} \log\left[\operatorname{Id} - \eta_k \nabla^2 \mathcal{C}_s(h_k)\right].
\end{aligned}
\tag{9}
$$

where the last equality follows from the Jacobi formula for positive definite matrices, namely $\log \det = \operatorname{tr} \log$.

We now use the same Markov argument as in Theorem 1, with posterior $\widetilde{\rho}_K^s$ and prior $\rho_0$. Explicitly, we have that with probability at least $1 - \frac{\delta}{2}$ on $(s, h_K) \sim \mu^m * \widetilde{\rho}_K$

$$e^{m\Psi(\mathcal{L}_s(h_K), \mathcal{L}_{\mathcal{Z}}(h_K)) - \log \frac{\widetilde{\rho}_K^s(h_K)}{\rho_0(h_K)}} \le \frac{2}{\delta} \int_{\mathcal{Z}^m \times \mathcal{H}} e^{m\Psi(\mathcal{L}_{\bar{s}}(h), \mathcal{L}_{\mathcal{Z}}(h)) - \log \frac{\widetilde{\rho}_K^{\bar{s}}(h)}{\rho_0(h)}} \mathrm{d}\mu^m(\bar{s}) \mathrm{d}\widetilde{\rho}_K^{\bar{s}}(h).$$

Since for every $\bar{s} \in \mathcal{Z}^m$ we have

$$
\begin{aligned}
\int_{\mathcal{H}} & e^{m\Psi(\mathcal{L}_{\bar{s}}(h), \mathcal{L}_{\mathcal{Z}}(h)) - \log \frac{\widetilde{\rho}_K^{\bar{s}}(h)}{\rho_0(h)}} \mathrm{d}\widetilde{\rho}_K^{\bar{s}}(h) \\
&= \int_{\{\widetilde{\rho}_k > 0\}} e^{m\Psi(\mathcal{L}_{\bar{s}}(h), \mathcal{L}_{\mathcal{Z}}(h)) - \log \frac{\widetilde{\rho}_K^{\bar{s}}(h)}{\rho_0(h)}} \mathrm{d}\widetilde{\rho}_K^{\bar{s}}(h) \le \int_{\mathcal{H}} e^{m\Psi(\mathcal{L}_{\bar{s}}(h), \mathcal{L}_{\mathcal{Z}}(h))} \mathrm{d}\rho_0(h),
\end{aligned}
$$

we get that

$$\mu^m * \rho_K \left( m\Psi\big(\mathcal{L}_s(h_K), \mathcal{L}_\mathcal{Z}(h_K)\big) \leq \log \frac{\widetilde{\rho}_K^s(h_K)}{\rho_0(h_K)} + \log \frac{2\xi}{\delta} \,\Big|\, h_0 \in A_s^0 \right) \geq 1 - \delta/2 \,.$$

Now, we note that for any $h_0 \in A_s^0$, the following holds:

$$\log \frac{\widetilde{\rho}_k^s(h_K)}{\rho_0(h_K)} = \log \frac{\widetilde{\rho}_k^s(h_K)}{\widetilde{\rho}_0^s(h_0)} + \log \frac{\rho_0^s(h_0)}{\rho_0(h_K)} - \log \rho_0(A_s^0) \,,$$

which is further bounded noticing that $-\log(1 - \delta/2) \leq \delta$ as $\delta \in (0, 1)$. In particular, using the change of density formula (9) we get that

$$\mu^m * \rho_K \bigg( m\Psi\big(\mathcal{L}_s(h_K), \mathcal{L}_\mathcal{Z}(h_K)\big)$$
$$\leq \log \frac{\rho_0(h_0)}{\rho_0(h_K)} - \sum_{k=0}^{K-1} \mathrm{tr} \log \Big[ \mathrm{Id} -\eta_k \nabla^2 \mathcal{C}_s(h_k) \Big] + \log \frac{2\xi}{\delta} + \delta \,\Big|\, h_0 \in A_s^0 \bigg) \geq 1 - \delta/2 \,.$$
(10)

Now, note that for any event $E$ we have

$$\mu^m * \rho_K(E) = \int_{\mathcal{Z}^m} \big( \rho_K(E|h_0 \in A_s^0)\rho_0(A_s^0) + \mu^m * \rho_K(E|h_0 \in A_s^0)(1 - \rho_0(A_s^0)) \big) \, \mathrm{d}\mu^m(s)$$
$$\leq \mu^m * \rho_K(E|h_0 \in A_s^0) + \frac{\delta}{2} \,,$$

since $\mu^m \otimes \rho_0(h_0 \notin A_0^s) \leq \delta/2$ by hypothesis. Applying this to (10), we get that

$$m\Psi\big(\mathcal{L}_s(h_K), \mathcal{L}_\mathcal{Z}(h_K)\big) \leq \log \frac{\rho_0(h_0)}{\rho_0(h_K)} - \sum_{k=0}^{K-1} \mathrm{tr} \log \Big[ \mathrm{Id} -\eta_k \nabla^2 \mathcal{C}_s(h_k) \Big] + \log \frac{2\xi}{\delta} + \delta \,, \quad (11)$$

with probability at least $1 - \delta$ on $(s, h_K) \sim \mu^m * \rho_K$. Since sampling from $\rho_K$ for a given $s$ is equivalent to sample from $\rho_0$ and follow the dynamics until the $K$-th step, we can claim that (11) holds with probability at least $1 - \delta$ on $(s, h_0) \sim \mu^m \otimes \rho_0$. ∎

**Lemma 5** *With the notation of Theorem 4, let $h_k \in A_s$. Then*

$$-\mathrm{tr} \log \Big( \mathrm{Id} -\eta_k \nabla^2 \mathcal{C}_s(h_k) \Big) \leq \eta_k \Delta \mathcal{C}_s(h_k) + \eta_k^2 \|\nabla^2 \mathcal{C}_s(h_k)\|_\mathrm{F}^2 \leq \frac{3}{2}\eta_k \|\nabla^2 \mathcal{C}_s(h_k)\|_\mathrm{TR},$$

*where $\| \cdot \|_\mathrm{F}$ is the Frobenius norm and $\| \cdot \|_\mathrm{TR}$ the trace norm.*

**Proof** To obtain the first upper bound, denote as $\{\lambda_i(h_k)\}_{i=1}^d$ the spectrum of $\nabla^2 \mathcal{C}_s(h_k)$. Then we have that

$$-\mathrm{tr} \log \Big( \mathrm{Id} -\eta_k \nabla^2 \mathcal{C}_s(h_k) \Big) = -\sum_{k=0}^{K-1} \sum_{i=1}^d \log(1 - \eta_k \lambda_i(h_k)) \,.$$

Using that $-\log(1-u) \le u(u+1)$ for $|u| \le 1/2$, we obtain that for each $k$

$$-\sum_{i=1}^{d} \log(1 - \eta_k \lambda_i(h_k)) \le \eta_k \sum_{i=1}^{d} \lambda_i(h_k) + \eta_k^2 \sum_{i=1}^{d} \lambda_i(h_k)^2 = \eta_k \Delta \mathcal{C}_s(h_k) + \eta_k^2 \|\nabla^2 \mathcal{C}_s(h_k)\|_{\mathrm{F}}^2 \,.$$

For the second upper bound, note that $\Delta \mathcal{C}_s(h_k) \le \|\nabla^2 \mathcal{C}_s(h_k)\|_{\mathrm{TR}}$ and

$$\eta_k^2 \|\nabla^2 \mathcal{C}_s(h_k)\|_{\mathrm{F}}^2 = \eta_k^2 \sum_{k=1}^{d} \lambda_i(h_k)^2 \le \eta_k \sum_{k=1}^{d} |\eta_k \lambda_i(h_k)||\lambda_i(h_k)| = \frac{1}{2} \eta_k \|\nabla^2 \mathcal{C}_s(h_k)\|_{\mathrm{TR}} \,,$$

where we used that $|\eta_k \lambda_i(h_k)| \le \eta_k \|\nabla^2 \mathcal{C}_s(h_k)\| \le 1/2$ since $\mathcal{C}_s$ is $1/(2\eta_k)$-smooth in $h_k \in A_s$. ∎

## Appendix B. A few more explicit bounds

In this section we leverage and extend the approach of Casado et al. (2024) (we show here that it is possible to drop the absolute continuity assumption for the loss required therein) to give a possible way to choose $\Psi$ (in Theorems 1 and 4) so that the term $\xi$ appearing the the bounds can be explicitly controlled. For the sake of conciseness we will state everything in the continuous time framework, as the discrete time counterpart of these results will naturally follow replacing the use of Theorem 1 with Theorem 4.

For $b \in (0, +\infty]$, let $\psi : [0, b) \to [0, +\infty)$ be a convex, continuously differentiable function, such that $\psi(0) = \psi'(0) = 0$. We define

$$B = \lim_{\lambda \to b} \frac{\psi(\lambda)}{\lambda}$$

and

$$A = \begin{cases} \lim_{\lambda \to b} (\lambda B - \psi(\lambda)) & \text{if } B < +\infty; \\ +\infty & \text{otherwise.} \end{cases}$$

For $u \in \mathbb{R}$, we define the Fenchel conjugate of $\psi$ as

$$\psi^{\star}(u) = \sup_{\lambda \in [0,b)} (\lambda u - \psi(\lambda)) \,.$$

The following lemma summarises a few properties of $\psi^{\star}$. Its claims are either immediate to check or classical results for convex functions (see, for instance, Chapter 2 in Boucheron et al. 2013).

**Lemma 10** $\psi^{\star}$ *satisfies the following properties:*

- $\psi^{\star}(u) = 0$ *if* $u \le 0$;

- $\psi^{\star}(u) = +\infty$ *if* $u > B$ *and* $b = +\infty$;

- $\psi^{\star}(u) = A + b(u - B)$ *if* $u > B$ *and* $b < +\infty$;

- *the restriction of* $\psi^{\star}$ *to* $[0, B]$ *is a continuously differentiable bijection* $[0, B] \leftrightarrow [0, A]$.

*Moreover, we have that for $v \in [0, A]$*

$$\psi^{\star-1}(v) = \inf_{\lambda \in (0,b)} \frac{v + \psi(\lambda)}{\lambda} \in [0, B]$$

*and, if $b < +\infty$, $\psi^\star$ is invertible on $[0, +\infty)$ and for $v > A$*

$$\psi^{\star-1}(v) = \frac{v - A}{b} + B.$$

Given a loss function $\ell$, we will say that $\ell$ has sub-$\psi$ concentration if, for all $h \in \mathcal{H}$, it satisfies

$$\log \int e^{\lambda(\mathcal{L}_{\mathcal{Z}}(h) - \ell(h,z))} \mathrm{d}\mu(z) \leq \psi(\lambda),$$

for all $\lambda \in [0, b)$. As an explicit example, we note that for $\psi(\lambda) = R^2\lambda^2/2$ defined on $[0, \infty)$, the sub-$\psi$ concentration follows from $R$-sub-Gaussianity.

**Lemma 11** *If $\ell$ has sub-$\psi$ concentration, then for each $h$ we have that for any $u \in \mathbb{R}$*

$$\mu^m\big(\mathcal{L}_{\mathcal{Z}}(h) \geq \mathcal{L}_s(h) + u\big) \leq e^{-m\psi^\star(u)}.$$

**Proof** For $u < 0$ the result is trivial, as the RHS is 1. For $u \geq 0$, it follows from a simple application of Markov's inequality, since we have that

$$
\begin{aligned}
\mu^m\big(\mathcal{L}_{\mathcal{Z}}(h) \geq \mathcal{L}_s(h) + u\big) &= \mu^m\left(\sum_{z \in s}\big(\mathcal{L}_{\mathcal{Z}}(h) - \ell(h, z)\big) \geq mu\right) \\
&= \inf_{\lambda \in [0,b)} \mu^m\left(e^{\lambda \sum_{z \in s}(\mathcal{L}_{\mathcal{Z}}(h) - \ell(h,z))} \geq e^{m\lambda u}\right) \\
&\leq \inf_{\lambda \in [0,b)} e^{-m\lambda u} \prod_{z \in s} \int_{\mathcal{Z}} e^{\lambda(\mathcal{L}_{\mathcal{Z}}(h) - \ell(z,h))} \mathrm{d}\mu(z) = e^{-m\psi^\star(u)},
\end{aligned}
$$

which is what we wanted. ∎

**Corollary 12** *If $\ell$ has sub-$\psi$ concentration, then for each $h$, for any $v \in \mathbb{R}$*

$$\mu^m\big(m\psi^\star(\mathcal{L}_{\mathcal{Z}}(h) - \mathcal{L}_s(h)) \geq v\big) \leq \min\{1, e^{-v}\}.$$

**Proof** First, note that the result if trivially true if $v \leq 0$, as the RHS is 1. For $v > 0$, first notice that if $b < +\infty$ then $\psi^\star$ is non-decreasing and invertible on $[0, +\infty)$ by Lemma 10, and so

$$\mu^m\big(m\psi^\star(\mathcal{L}_{\mathcal{Z}}(h) - \mathcal{L}_s(h)) \geq v\big) = \mu^m\big(\mathcal{L}_{\mathcal{Z}}(h) - \mathcal{L}_s(h) \geq \psi^{\star-1}(v/m)\big).$$

In this case the conclusion follows immediatly by Lemma 11.

On the other hand, if $b = +\infty$, for $v > Am$, we have that $m\psi^\star(\mathcal{L}_{\mathcal{Z}}(h) - \mathcal{L}_s(h)) \geq v$ only if $\mathcal{L}_{\mathcal{Z}}(h) - \mathcal{L}_s(h) > B$. However, by Lemma 11 this is a zero-probability event. Indeed, we have that, for any $\varepsilon > 0$, $\mu^m\big(\mathcal{L}_{\mathcal{Z}}(h) \geq \mathcal{L}_s(h) + B + \varepsilon\big) \leq e^{-m\psi^\star(B+\varepsilon)} = 0$, since $\psi^\star(B + \varepsilon) = +\infty$ by Lemma 10. So, for $v > Am$ we have

$$\mu^m\big(m\psi^\star(\mathcal{L}_{\mathcal{Z}}(h) - \mathcal{L}_s(h)) \geq v\big) = 0 \leq e^{-v}.$$

We are only left with the case $v \in [0, Am]$ and $b = +\infty$. In such a case, the invertibility of $\psi^\star$ on $[0, B]$ is enough and we get that (almost surely)

$$\left\{m\psi^\star(\mathcal{L}_\mathcal{Z}(h) - \mathcal{L}_s(h)) \geq v\right\} = \left\{\mathcal{L}_\mathcal{Z}(h) - \mathcal{L}_s(h) \geq \psi^{\star-1}(v/m)\right\},$$

and so again

$$\mu^m\big(m\psi^\star(\mathcal{L}_\mathcal{Z}(h) - \mathcal{L}_s(h)) \geq v\big) = \mu^m\big(\mathcal{L}_\mathcal{Z}(h) - \mathcal{L}_s(h) \geq \psi^{\star-1}(v/m)\big).$$

Again, we conclude by Lemma 11, since $\psi^\star$ is a bijection $[0, B] \leftrightarrow [0, A]$ by Lemma 10. ∎

**Lemma 13** *If $\ell$ has sub-$\psi$ concentration, for any $m' < m$, for any fixed $h$, we have*

$$\int_{\mathcal{Z}^m} e^{m'\psi^\star(\mathcal{L}_\mathcal{Z}(h) - \mathcal{L}_s(h))} \mathrm{d}\mu^m(s) \leq \frac{m}{m - m'}.$$

**Proof** We have

$$\int_{\mathcal{Z}^m} e^{m'\psi^\star(\mathcal{L}_\mathcal{Z}(h) - \mathcal{L}_s(h))} \mathrm{d}\mu^m(s) = \int_0^\infty \mu^m\left(e^{m'\psi^\star(\mathcal{L}_\mathcal{Z}(h) - \mathcal{L}_s(h))} \geq u\right) \mathrm{d}u$$

$$= \int_0^\infty \mu^m\left(m\psi^\star(\mathcal{L}_\mathcal{Z}(h) - \mathcal{L}_s(h)) \geq \frac{m}{m'}\log u\right)\mathrm{d}u \leq \int_0^\infty \min(1, e^{-\frac{m}{m'}\log u})\mathrm{d}u$$

$$= 1 + \int_1^\infty e^{-\frac{m}{m'}\log u}\mathrm{d}u = \frac{m}{m - m'},$$

where we applied Lemma 12 to obtain the inequality. ∎

We can now state the following generalisation bound, which follows from Theorem 1.

**Proposition 14** *Consider the dynamics $\partial_t h_t = -\nabla \mathcal{C}_s(h_t)$, with $\mathcal{C}_s : \mathcal{H} \to \mathbb{R}$ twice differentiable. Assume that $\ell$ has sub-$\psi$ concentration. Fix $T > 0$ and $\delta \in (0, 1)$. We have that, with probability at least than $1 - \delta$ on the draw of the pair $(s, h)$ from $\mu^m \otimes \rho_0$,*

$$\mathcal{L}_\mathcal{Z}(h_T) \leq \mathcal{L}_s(h_t) + \inf_{\lambda \in (0,b)} \left(\frac{1}{\lambda(m-1)}\left(\log\frac{\rho_0(h_0)}{\rho_0(h_T)} + \int_0^T \Delta\mathcal{C}_s(h_t)\mathrm{d}t + \log\frac{m}{\delta}\right) + \frac{\psi(\lambda)}{\lambda}\right). \tag{12}$$

**Proof** We apply Theorem 1 with the choice

$$\Psi(u, v) = \frac{m-1}{m}\psi^\star(v - u).$$

This leads to the high probability bound

$$\psi^\star\big(\mathcal{L}_\mathcal{Z}(h_T) - \mathcal{L}_s(h_T)\big) \leq \frac{1}{m-1}\left(\log\frac{\rho_0(h_0)}{\rho_0(h_T)} + \int_0^T \Delta\mathcal{C}_s(h_t)\mathrm{d}t + \log\frac{\xi}{\delta}\right),$$

with

$$\xi = \int_{\mathcal{Z}^m \times \mathcal{H}} e^{(m-1)\Psi(\mathcal{L}_{\bar{s}}(h), \mathcal{L}_\mathcal{Z}(h))} \mathrm{d}\mu^m(\bar{s})\mathrm{d}\rho_0(h).$$

Applying Corollary 12 with $m' = m - 1$ we get that $\xi \leq \log m$, and the conclusion follows from inverting $\psi^\star$ as in Lemma 10. ∎

As a consequence of the result above, we can get a bound for sub-exponential losses. In particular, we recall that $\ell(h, \cdot)$ is $(R, \alpha)$-sub-exponential if for any $\lambda \in (-\alpha^{-1}, \alpha^{-1})$

$$\log \int_{\mathcal{Z}} e^{\lambda \ell(h,z)} \mathrm{d}\mu(z) \leq \frac{R^2 \lambda^2}{2} .$$

**Corollary 15** *Assume that, for each $h \in \mathcal{H}$, $\ell(h, \cdot)$ is $(R, \alpha)$-sub-exponential. Then, for any $\delta \in (0, 1)$ and $T > 0$, with probability at least $1 - \delta$ on the random draw $(s, h_0) \sim \mu^m \otimes \rho_0$, we have*

$$\mathcal{L}_{\mathcal{Z}}(h_T) \leq \mathcal{L}_s(h_T) + \inf_{\lambda \in (0, \alpha^{-1})} \left( \frac{1}{\lambda(m-1)} \left( \log \frac{\rho_0(h_0)}{\rho_0(h_T)} + \int_0^T \Delta \mathcal{C}_s(h_t) \mathrm{d}t + \log \frac{m}{\delta} \right) + \frac{R^2 \lambda}{2} \right) .$$

**Proof** We have that $\ell$ has $\psi$-concentration, with $\psi(\lambda) = R^2 \lambda^2 / 2$ and $b = 1/\alpha$. Hence, the result follows directly from Proposition 14. ∎

**Corollary 16** *Assume that, for each $h \in \mathcal{H}$, $\ell(h, \cdot)$ is $R$-sub-Gaussian. Then, for any $\delta \in (0, 1)$ and $T > 0$, with probability at least $1 - \delta$ on the random draw $(s, h_0) \sim \mu^m \otimes \rho_0$, we have*

$$\mathcal{L}_{\mathcal{Z}}(h_T) \leq \mathcal{L}_s(h_T) + R \sqrt{\frac{2 \left( \log \frac{\rho_0(h_0)}{\rho_0(h_T)} + \int_0^T \Delta \mathcal{C}_s(h_t) \mathrm{d}t + \log \frac{m}{\delta} \right)}{(m-1)}} .$$

**Proof** As $\ell(h, \cdot)$ has $\psi$-concentration, with $\psi(\lambda) = R^2 \lambda^2 / 2$ and $b = \infty$, we can apply Proposition 14. Setting

$$\lambda = \sqrt{\frac{2 \left( \log \frac{\rho_0(h_0)}{\rho_0(h_T)} + \int_0^T \Delta \mathcal{C}_s(h_t) \mathrm{d}t + \log \frac{m}{\delta} \right)}{R^2(m-1)}}$$

in (12) we obtain the desired result. ∎

We note that in the $R$-sub-Gaussian case, setting $\lambda = 1$ in (12) leads to the bound of Corollary 2, up to replacing $m$ with $m - 1$ and to neglect a term $\log m$. As a consequence, for large $m$ one can expect the bound of Corollary 16 to be tighter than the one in Corollary 2.

## Appendix C. Examples

### C.1. Random feature model

We recall from Section 5.1 that $\mathcal{X} = S^{p-1}$ and $\mathcal{Y} = [0, 1]$. The goal is to learn a target function $f : \mathcal{X} \to \mathcal{Y}$, with $F_h(x) = \frac{1}{\sqrt{d}} h \cdot \Phi(x)$ and $f_h = \min\{0, \max\{1, F_h\}\}$. We consider a quadratic loss $\ell(h, z) = (f_h(x) - y)^2$ and an optimisation objective $\mathcal{C}_s(h) = \frac{1}{m} \sum_{z \in s} \hat{\ell}(F_h(x), y)$, with $\hat{\ell}(F, y) = (F - y)^2$. The features are given by $\Phi(x) = \phi(Wx)$, where $\phi$ is a non-linearity acting component-wise, and $W$ is a $d \times p$ matrix whose components are independently drawn from a standard Gaussian distribution.

**Proposition 6** *For the random feature model described above, we have the limit in probability*

$$\lim_{d \to \infty} \left( \log \frac{\rho_0(h_0)}{\rho_0(h_T)} + \int_0^T \Delta \mathcal{C}_s(h_t) \mathrm{d}t \right) \leq T \left( \mathbb{E}_{\zeta \sim \mathcal{N}(0,1)}[\phi(\zeta)^2] + 2\sqrt{\mathcal{C}_s(h_0)} \right).$$

**Proof** A simple calculation shows that $\Delta \mathcal{C}_s(h) = \frac{2}{m} \sum_{z \in s} \frac{1}{d} \|\Phi(x)\|^2$. As $d \to \infty$ we get

$$\frac{1}{d}\|\Phi(x)\|^2 = \frac{1}{d} \sum_{i=1}^d \phi(W_i \cdot x)^2 \to \mathbb{E}[\phi(W_1 \cdot x)^2] = \mathbb{E}_{\zeta \sim \mathcal{N}(0,1)}[\phi(\zeta)^2],$$

where $W_i$ denotes the $i$-th raw of $W$. Hence,

$$\int_0^T \Delta \mathcal{C}_s(h_t) \mathrm{d}t = T \mathbb{E}_{\zeta \sim \mathcal{N}(0,1)}[\phi(\zeta)^2].$$

On the other hand, recalling that under $\rho_0$ all the components of $h$ are independently distributed as standard Gaussians, we get

$$\log \frac{\rho_0(h_0)}{\rho_0(h_T)} = \frac{1}{2} \left( \|h_T\|^2 - \|h_0\|^2 \right) = \int_0^T h_t \cdot \partial_t h_t \, \mathrm{d}t = -\int_0^T h_t \cdot \nabla \mathcal{C}_s(h_t) \, \mathrm{d}t.$$

Our choice of quadratic learning objective yields

$$-h_t \cdot \nabla \mathcal{C}_s(h_t) = \frac{2}{m} \sum_{z \in s} (y - F_{h_t}(x)) \frac{\Phi(x) \cdot h_t}{\sqrt{d}} = \frac{2}{m} \sum_{z \in s} (y - F_{h_t}(x)) F_{h_t}(x)$$

$$= -2\mathcal{C}_s(h_t) + \frac{2}{m} \sum_{z \in s} y(y - F_{h_t}(x)) \leq \frac{2}{m} \sum_{z \in s} |F_{h_t}(x) - y| \leq 2\sqrt{\mathcal{C}_s(h_t)},$$

where we used that $|y| = |f(x)| \leq 1$ and Cauchy-Schwarz inequality. The training dynamics ensure that $\mathcal{C}_s(h_t) \leq \mathcal{C}_s(h_0)$. ∎

**Proposition 7** *Let $f : S^{d-1} \to [0, 1]$ be a linear combination of spherical harmonics of degree at most $D$. Let $\phi$ be given by $\phi(x) = \sum_{k=1}^J (x/e)^j$, with $J \geq D$. Then, in the overparameterised regime $d \to \infty$, there is a $\lambda > 0$, independent of $m$ and on the training dataset $s$, such that for $T = \frac{\log m}{4\lambda}$ we have $\mathcal{L}_{\mathcal{Z}}(h_T) \leq O\left( \frac{\log m}{m} \right)$, asymptotically for large $m$.*

**Proof** Here we assume that $d$ is large enough with respect to $m$, so that, up to negligible correction we can approximate $\frac{1}{d}\Phi(x) \cdot \Phi(x')$ by its overparameterised limit $(d \to \infty)$, given by (Daniely et al., 2017)

$$\Theta(x, x') = \mathbb{E}_{(\xi, \xi') \sim \mathcal{N}(0, \Sigma)}[\phi(\xi)\phi(\xi')].$$

By Lemma 17, we have that

$$\Theta(x, x') = \sum_{j=1}^J \sum_{l=1}^{N_j} \lambda_j \psi_{l,j}(x) \psi_{l,j}(x'),$$

where $\psi_{l,j}$'s denote the spherical harmonics of degree $l$, and $N_l$ is the number of spherical harmonics of degree $l$.

Let us define the $m \times m$ renormalised Gram matrix $\hat{\Theta} = \frac{1}{m}\big(\Theta(x, x')\big)_{(z,z') \in s \times s}$. If we denote as $V_{l,j}$ the $m$-vector $\big(\psi_{l,j}(x)\big)_{x \in s}$, we have that

$$\hat{\Theta} = \frac{1}{m} \sum_{j=1}^{J} \sum_{l=1}^{N_j} \lambda_j V_{l,j} V_{l,j}^{\top} \,.$$

In particular, $\hat{\Theta}$ has rank at most $\sum_{j=1}^{J} N_j$, and its eigendecomposition can be written as

$$\hat{\Theta} = \sum_{j=1}^{J} \sum_{l=1}^{N_j} \hat{\lambda}_{l,j} \hat{V}_{l,j} \hat{V}_{l,j}^{\top}$$

for some orthonormal vectors $\hat{V}$'s. It is known that for large $m$, the eigenvalues of the Gram matrix tend to those of the kernel. More precisely, from Shawe-Taylor et al. (2005), we have that with probability higher than $1 - \tilde{\delta}$, uniformly on $j$ and $l$

$$\hat{\lambda}_{l,j} \geq \lambda_j - O\left(\sqrt{\frac{\log \sum_{j=1}^{J} N_j - \log \tilde{\delta}}{m}}\right) \,.$$

We can choose a $\lambda > 0$ such that $\lambda_j \geq 2\lambda$, for all $j$. Then, fixed $\tilde{\delta}$, if $m$ is large enough we have that for all $j$ and all $l$, $\lambda_{l,j} \geq \lambda > 0$ with high probability. In particular, when this is true, we have that $\hat{\Theta}$ has rank $\sum_{j=1}^{J} N_j$, and so all the $V_{l,j}$ are independent.

Let $\Delta_t$ be the $m$-vector $(F_{h_t}(x) - y)_{z \in s}$. The dynamics now reads

$$\partial_t \Delta_t = -2\hat{\Theta}\Delta_t \,,$$

and so we get

$$\Delta_t = e^{-2t\hat{\Theta}} \Delta_0 \,.$$

Now, let us show that $\Delta_0$ lies in the range of $\hat{\Theta}$. Indeed, we have that the network's output at initialisation behaves as a Gaussian process (Neal, 1995). More precisely (Grenander, 1950), we have that

$$F_{h_0}(x) = \sum_{j=1}^{J} \sum_{l=1}^{N_j} \sqrt{\lambda_j} \zeta_{l,j} \psi_{l,j}(x) \,,$$

where the $\zeta$'s are i.i.d. draws from a standard Gaussian. In particular, we know that $F_{h_0}(x)$ lies in the span of the spherical harmonics of degree at most $J$. Since we have that $f$ also lies in this span by assumption, we can write $\Delta_0$ as a linear combination of the $V_{l,j}$'s. Since the $V_{l,j}$'s are linearly independent, we have that each of them is orthogonal to the ker of $\hat{\Theta}$. Indeed, let $w \in \mathrm{ker}\hat{\Theta}$. We have

$$0 = \hat{\Theta}w = \frac{1}{m} \sum_{j=1}^{J} \sum_{l=1}^{N_j} \lambda_j (w \cdot V_{l,j}) V_{l,j} \,.$$

Since the $V_{l,j}$'s are linearly independent, and all the $\lambda_j$ are non-zero, we have that $(w \cdot V_{l,j})$ must be zero for all $l$ and all $j$, which means that each $V_{l,j}$ is orthogonal to the ker of $\hat{\Theta}$, and hence is in its range. From this it follows that $\Delta_0$ is in the range of $\hat{\Theta}$.

Now, we can write

$$\mathcal{C}_s(h_t) = \frac{1}{m}\|\Delta_t\|^2 \leq e^{-4\lambda t}\frac{1}{m}\|\Delta_0\|^2 = e^{-4\lambda t}\mathcal{C}_s(h_0) \,,$$

where we used the fact that $\Delta_0$ lies in the range of $\hat{\Theta}$, where the smallest eigenvalue of $\hat{\Theta}$ is lower-bounded by $\lambda$. Choosing $T = \frac{\log m}{4\lambda}$, we get that

$$\mathcal{C}_s(h_T) \leq \frac{\mathcal{C}_s(h_0)}{m} \,.$$

Now, fixed any $\hat{\delta}$, we can find a constant $C$ such that with probability higher than $1 - \hat{\delta}$,

$$\sup_{x \in \mathcal{X}}(F_{h_0}(x) - f(x)) \leq U \,.$$

In particular, we have that

$$\mathcal{L}_s(h_T) \leq \mathcal{C}_s(h_T) = O(1/m) \,,$$

with probability higher than $1 - \tilde{\delta} - \hat{\delta}$. From Proposition 6 and (4) it follows that, with probability higher than $1 - \delta - \tilde{\delta} - \hat{\delta}$

$$\mathcal{L}_{\mathcal{Z}}(h_T) = O\left(\frac{\log m}{m}\right) \,,$$

which is what we wanted to show. ∎

**Lemma 17** *Let*

$$\phi(x) = \sum_{j=0}^{J} \gamma^j x^j$$

*where $\gamma \in \mathbb{R} \setminus \mathbb{Q}$. The we have that the range of $\Theta$ coincides with the span of the spherical harmonics of degree at most $J$.*

**Proof** Since $\phi$ is a polynomial of order $J$, we have that it can be written as a linear combination of the first $J + 1$ Hermite polynomials (up to order $J$). We hence have

$$\phi(x) = \sum_{j=0}^{J} \alpha_j P_j(x) \,.$$

Note that all the Hermite polynomials have rational coefficients. Since $\gamma$ is irrational, $\gamma^j$ cannot be written as a finite linear combination of powers $\gamma^{j'}$ (with $j' \neq j$) with rational coefficient. In particular, we have that

$$\alpha_J = \hat{\alpha}_{J,J}\gamma^J \,;$$
$$\alpha_{J-1} = \hat{\alpha}_{J-1,J}\gamma^J + \hat{\alpha}_{J-1,J-1}\gamma^{J-1} \,;$$
$$\cdots$$
$$\alpha_j = \sum_{J \geq k \geq j} \hat{\alpha}_{j,k}\gamma^k \,,$$

for some rational non-null coefficients $\hat{\alpha}$'s. From Daniely et al. (2017), we know that we can write $\Theta(x \cdot x') = R(x \cdot x')$, where $R$ is a polynomial given by

$$R(u) = \sum_{j=0}^{J} \alpha_j^2 u^j \,.$$

We can write

$$\alpha_J^2 = \tilde{\alpha}_{J,2J} \gamma^{2J} \,;$$
$$\alpha_{J-1}^2 = \tilde{\alpha}_{J-1,2J} \gamma^{2J} + \tilde{\alpha}_{J-1,2J-1} \gamma^{2J-1} + \tilde{\alpha}_{J-1,2J-2} \gamma^{2J-2} \,;$$
$$\cdots$$
$$\alpha_j^2 = \sum_{2J \geq k \geq 2j} \tilde{\alpha}_{j,k} \gamma^k \,,$$

where again the coefficients $\tilde{\alpha}$'s are rational numbers. Note moreover that $\tilde{\alpha}_{j,2j}$ is always non-zero. Rewriting $R$ as a combination of Gegenbauer polynomials of index $\frac{p-2}{2}$ (see, *e.g.*, Yang and Salman, 2020), we have

$$R(u) = \sum_{j=0}^{J} \beta_j Q_j(u) \,,$$

where $Q_j = C_j^{\left(\frac{p-2}{2}\right)}$ is the Gegenbauer polynomial of degree $j$. These polynomial also have only rational coefficients. Since none of the $\alpha_j^2$'s can be written as a linear combination (with rational coefficients) of the others, it must be that none of the $\beta_j$'s is zero. From Theorem H.12 in Yang and Salman (2020), it follows that $\Theta$ has $J$ non-zero eigenvalues, given by

$$\lambda_j = \frac{p-2}{p+2j-2} \beta_j$$

and that the eigenspace associated to $\lambda_j$ is of the spherical harmonics of degree $t$. ∎

**Remark 18 (Upper bounding the Laplacian for the cross entropy loss)** *We remark here that we used the square loss for the training of the model because it leads to the simple closed form expression for the log density ratio in Proposition 6. If we had considered a multiclass classification task, a more natural loss function would have been the cross entropy loss. Although in such a case it is not clear if a simple upper bound can always be stated for the log density ratio, it is the case that the Laplacian term can be upper bounded easily.*

*We let*

$$F(x) = \frac{1}{\sqrt{d}} h \Phi(x) \,,$$

*where $h$ is now a $q \times d$ learnable matrix and $\Phi : \mathbb{R}^p \to \mathbb{R}^d$ is a fixed feature map, chosen randomly at initialisation.*

*In general, for a generic model with a twice differentiable $\hat{\ell}$ one has that*

$$\Delta \hat{\ell} = \sum_i \frac{\partial \hat{\ell}}{\partial F^i} \Delta F^i + \sum_{ii'} \frac{\partial^2 \hat{\ell}}{\partial F^i \partial F^{i'}} \nabla F^i \cdot \nabla F^{i'} \,,$$

*which follows easily from the chain rule. Here the model is linear in $h$ and so we are simply left with*

$$\Delta \hat{\ell} = \frac{1}{d} \|\Phi\|^2 \Delta_F \hat{\ell} \,,$$

*where $\Delta_F \hat{\ell} = \sum_i \partial^2_{F^i} \hat{\ell}$.*

*When $\hat{\ell}$ is the cross entropy loss, we get*

$$\Delta_F \hat{\ell} = \sum_i \partial^2_{F^i} \hat{\ell} = \sum_i \left( \frac{e^{F^i}}{\sum_j e^{F^j}} - \frac{e^{2F^i}}{\left(\sum_j e^{F^j}\right)^2} \right) = 1 - \frac{\sum_i e^{2F^i}}{\left(\sum_i e^{F^i}\right)^2} \leq 1 \,,$$

*and so*

$$\Delta \mathcal{C}_s(h) = \frac{1}{m} \sum_{z \in s} \frac{1}{d} \|\Phi(x)\|^2 \left( 1 - \frac{\sum_i e^{2F^i(x)}}{\left(\sum_i e^{F^i(x)}\right)^2} \right) \leq \frac{1}{m} \sum_{z \in s} \frac{1}{d} \|\Phi(x)\|^2 \,.$$

*As a final remark, we note that if we consider discretised dynamics, since the objective is a convex function of the parameters, we have that*

$$\|\nabla^2 \mathcal{C}_s(h)\|_{\mathrm{TR}} = \Delta \mathcal{C}_s(h) \leq \frac{1}{m} \sum_{z \in s} \frac{1}{d} \|\Phi(x)\|^2 \,.$$

*This quantity stays finite as $d \to \infty$ (with arbitrarily high probability), showing that the naive bound $\eta_k \|\nabla^2 \mathcal{C}_s(h_k)\|_{\mathrm{TR}} = O(d)$ stated at end of Section 4 can be very loose in practice.*

## C.2. Wide Neural Networks

We consider a fully connected neural network $F_h : \mathbb{R}^{n_0} \to \mathbb{R}$, for some $n_0 \in \mathbb{N}$. For simplicity, we require that each hidden layer has the same width, $n \in \mathbb{N}$, and the same activation function, $\phi : \mathbb{R} \to \mathbb{R}$. We assume that the inputs are coming from a compact set $\mathcal{X} \subset \mathbb{R}^{n_0}$.

The network output is determined by an input $x^0 \in \mathcal{X}$, weights $\{W^l\}_{l=2}^L \subset \mathbb{R}^{n \times n}$, $W^1 \in \mathbb{R}^{n_0 \times n}$ and $W^{L+1} \in \mathbb{R}^{n \times 1}$, and biases $\{b^l\}_{l=1}^L \subset \mathbb{R}^n$ and $b^{L+1} \in \mathbb{R}$. We use $h$ to denote the vector of all weights and biases. The network's output is $F_h(x^0) = f_h^{L+1}(x^0)$ where we define the functions $f_h^l$ by,

$$f_h^0(x) = x, \quad f_h^{l+1}(x) = \phi(f_h^l(x))W^{l+1} + b^{l+1}, \quad \text{for } l = 0, \ldots, L,$$

where $\phi$ is applied component-wise. We consider a dataset $s = (x_i, y_i)_{i=1}^m \in \mathcal{Z}^m$ sampled from the measure $\mu^m$. We consider a square loss objective in the form

$$\mathcal{C}_s(h) = \frac{1}{m} \sum_{i=1}^m (F_h(x_i) - y_i)^2 \,.$$

We consider a Gaussian initialisation $\rho_0$, where the all of the parameters are independently drawn as

$$W_{ij}^l \sim \mathcal{N}\left(0, \frac{\sigma_w^2}{n}\right), \quad b_i^l \sim \mathcal{N}(0, \sigma_b^2) \,,$$

for some positive $\sigma_w$ and $\sigma_b$.

If the training step-size is scaled appropriately, the large width limit is known to reduce to the neural tangent kernel (NTK) dynamics (Jacot et al., 2018). Given $x, x' \in \mathcal{X}$, the value of the NTK $\Theta(x, x') \in \mathbb{R}$ is given by the limit (in probability) as $n \to \infty$ of the quantity

$$\hat{\Theta}(x, x') = \frac{1}{n} \langle \nabla F_{h_0}(x), \nabla F_{h_0}(x') \rangle,$$

with $h_0 \sim \rho_0$.

We borrow the analysis of Lee et al. (2020), who use this fact to study the convergence on the finite width NN under training with gradient descent. To leverage ideas from this analysis, we must make the following additional assumptions:

1. The analytic NTK $\Theta$ is full-rank with minimum and maximum eigenvalues satisfying $0 < \lambda_{\min} \leq \lambda_{\max} < \infty$.

2. $\mathcal{Z} = \mathcal{X} \times \mathcal{Y}$ is compact and the data distribution $\mu$ has no atoms.

3. The activation function $\phi$ has Lipschitz continuous and bounded gradients.

Throughout this section, we use the notation $\eta_\star = 2(\lambda_{\min} + \lambda_{\max})^{-1}$. We define the functions $(g_h^l)_{l=1}^{L+1}$ by $g_h^l(x) = g_h^{l+1}(\phi(x)W^{l+1} + b^{l+1})$ for $l = 0, ..., L$ such that $F_h = g_h^l \circ f_h^l$ for each $l = 0, ..., L+1$. We also define $\delta_h^l(x) = (\nabla g_h^l) \circ f_h^l(x)$.

**Lemma 19 (Lemma 1, Lee et al. (2020))** *Suppose assumptions 1-3 are satisfied, then there exists a constant $R > 0$ such that, for any fixed $C > 0$ and $\delta \in (0, 1)$, for $n$ sufficiently large we can find a convex subset $A(C, \delta, n) \subseteq \mathcal{H}$, such that $\rho_0(A(C, \delta, n)) \geq 1 - \delta/2$ and*

$$\|\nabla F_h(x) - \nabla F_{h'}(x)\| \leq \sqrt{n}R\|h - h'\|, \qquad \|\nabla F_h(x)\| \leq \sqrt{n}R, \quad and$$
$$\|\delta_h^l(x) - \delta_{h'}^l(x)\| \leq R_1\|h - h'\|, \qquad \|f_h^l(x) - f_{h'}^l(x)\| \leq R_1\sqrt{n},$$
$$\|\delta_h^l(x)\| \leq R_1, \qquad \|f_h^l(x)\| \leq R_1\sqrt{n}, \quad for \ l = 0, ..., L,$$

*for all $h, h' \in B(h_0, Cn^{-1/2})$, $h_0 \in A(C, \delta, n)$ and $x \in \mathcal{X}$.*

Here we use the notation $B(h, r)$ to denote the ball about $h$ of radius $r$. As usual, the notation $\nabla$ denote derivatives with respect to the parameters. The result is not stated so explicitly in the paper of Lee et al. (2020), but can be deduced easily by following the proof therein. The convexity of the set $A(C, \delta, n)$ follows from its construction by upper bounding the operator norms of the weight matrices. Similarly, we state more formally another result from this work.

**Lemma 20 (Theorem G.1, Lee et al. (2020))** *Suppose assumptions 1-3 are satisfied and $s \in \text{supp}(\mu^m)$. Then there exists constants $C, R_0 > 0$ such that for sufficiently large $n$, whenever $\sup_k \eta_k < \eta_\star/n$ and $h_0 \in A(C, \delta, n)$,*

$$\sum_{j=1}^k \|h_{j+1} - h_j\| \leq Cn^{-1/2}, \quad \mathcal{C}_s(h_k) \leq \exp\left(-\frac{\lambda_{\min}}{3}\sum_{k'=0}^{k-1}\bar{\eta}_{k'}\right)R_0, \quad for \ each \ k \in \mathbb{N}.$$

To control the change in magnitude of the parameters we define an additional function:

$$J_s(h)^2 = \frac{1}{m} \sum_{i=1}^{m} |\langle h, \nabla_h F_h(x_i) \rangle|^2. \tag{13}$$

In Remark 22, we discuss how this can be controlled.

**Lemma 21** *Suppose assumptions 1-3 are satisfied and $s \in \text{supp}(\mu^m)$. Then there exists constants $C, R_0 > 0$ such that for sufficiently large $n$, whenever $\sup_k \bar{\eta}_k < \eta_\star$ and $h_0 \in A(C, \delta, n)$,*

$$\|h_K\|^2 - \|h_0\|^2 \leq \frac{2R_3}{\lambda_{\min} n}(J_s(h_0) + R_2).$$

**Proof** From the mean value theorem, it follows that for each $k$ there exists a vector

$$\tilde{h}_k \in \{\alpha h_k + (1-\alpha)h_{k+1} : \alpha \in [0,1]\}$$

such that the following holds:

$$\begin{aligned}
\|h_{k+1}\|^2 - \|h_k\|^2 &= 2\tilde{h}_k^T(h_{k+1} - h_k) \\
&= -2\eta_k \langle \tilde{h}_k, \nabla \mathcal{C}_s(h_k) \rangle \\
&\leq -2\eta_k \langle h_k, \nabla \mathcal{C}_s(h_k) \rangle + 2\eta_k \|\tilde{h}_k - h_k\| \|\nabla \mathcal{C}_s(h_k)\|. \tag{14}
\end{aligned}$$

To bound the first term, we first recognise that for any $l = 1, ..., L+1$, we have $\nabla_{W_{ij}^l} F_h = (\partial_{W_{ij}^l} g_h^l) \circ f_h^l \cdot \phi(f^{l-1})$ and $\nabla_{b_i^l} F_h = (\partial_{b_i^l} g_h^l) \circ f_h^l$. Thus, we obtain,

$$\langle h, \nabla \mathcal{C}_s(h) \rangle = \frac{2}{m} \sum_{i=1}^{m} \left( (F_h(x_i) - y_i) \sum_{l=1}^{L} \langle \delta_h^l(x_i), f_h^l(x_i) \rangle \right).$$

With this, we more easily control the change in $\langle h_k, \nabla \mathcal{C}_s(h_k) \rangle$ over the iterations. From Lemma 19, it follows that,

$$\begin{aligned}
\Big| \langle \delta_{h_k}^l(x_i), f_{h_k}^l(x_i) \rangle &- \langle \delta_{h_0}^l(x_i), f_{h_0}^l(x_i) \rangle \Big| \\
&\leq \|\delta_{h_k}^l(x_i) - \delta_{h_0}^l(x_i)\| \|f_{h_k}^l(x_i)\| + \|\delta_{h_0}^l(x_i)\| \|f_{h_k}^l(x_i) - f_{h_0}^l(x_i)\| \leq 2R_1 C.
\end{aligned}$$

Thus, from the Cauchy-Schwarz inequality we obtain that

$$\begin{aligned}
\langle h_k, \nabla \mathcal{C}_s(h_k) \rangle &\leq 2\mathcal{C}_s(h_k)^{1/2} \sqrt{\frac{1}{m} \sum_{i=1}^{m} \left( \sum_{l=1}^{L} \langle \delta_{h_k}^l(x_i), f_{h_k}^l(x_i) \rangle \right)^2} \\
&\leq 2\mathcal{C}_s(h_k)^{1/2} \left( \sqrt{\frac{1}{m} \sum_{i=1}^{m} \left( \sum_{l=1}^{L} \langle \delta_{h_0}^l(x_i), f_{h_0}^l(x_i) \rangle \right)^2} + 2R_1 CL \right) \\
&= 2\mathcal{C}_s(h_k)^{1/2}(J_s(h_0) + 2R_1 CL).
\end{aligned}$$

To bound the second term of (14), we use both Lemma 19 and Lemma 20 to obtain

$$\|\tilde{h}_k - h_k\|\|\nabla \mathcal{C}_s(h_k)\| \leq 2\|h_{k+1} - h_k\|\mathcal{C}_s(h_k)^{1/2}\sqrt{\frac{1}{m}\sum_{i=1}^{m}\|\nabla F_{h_k}(x_i)\|^2}$$

$$\leq 2\mathcal{C}_s(h_k)^{1/2}CR.$$

With this, we obtain the bound,

$$\|h_{k+1}\|^2 - \|h_k\|^2 \leq 4\eta_k \mathcal{C}_s(h_k)^{1/2}\big(J^\star + CR + 2R_1CL\big). \tag{15}$$

Using Lemma 20, we first obtain the simple bound,

$$\sum_{k=0}^{K-1}\eta_k\mathcal{C}_s(h_k)^{1/2} \leq n^{-1}R_0^{1/2}\sum_{k=0}^{K}\bar{\eta}_k. \tag{16}$$

When $K$ is large, we use the contractions of the training loss to obtain a sharper bound:

$$\sum_{k=0}^{K-1}\eta_k\mathcal{C}_s(h_k)^{1/2} \leq n^{-1}\sum_{k=0}^{K-1}\bar{\eta}_k\exp\left(-\frac{\lambda_{\min}}{6}\sum_{k'=0}^{k-1}\bar{\eta}_{k'}\right)R_0^{1/2}$$

$$\leq n^{-1}\exp\left(\frac{\lambda_{\min}}{6}\bar{\eta}_{\max}\right)R_0^{1/2}\int_0^\infty \exp\left(-\frac{\lambda_{\min}}{6}t\right)dt$$

$$\leq \frac{6R_0^{1/2}}{\lambda_{\min}n}\exp\left(\frac{\lambda_{\min}}{6}\bar{\eta}_{\max}\right). \tag{17}$$

Thus, to obtain the final bound, we use the identity $\|h_K\|^2 - \|h_0\|^2 = \sum_{k=0}^{K-1}(\|h_{k+1}\|^2 - \|h_k\|^2)$, substituting (15) along with the bounds in (16) and (17). ∎

Now we state Proposition 8 more formally.

**Proposition 8 (Rigorous statement)** *Suppose assumptions 1-3 are satisfied, then there exists positive constants $C$, $R$, $R_0$, $R_2$, and $R_3$ such that, for any $\delta \in (0,1)$ and sufficiently large $n$, whenever $\sup_k \bar{\eta}_k < \min\{\eta_\star, (2R(R+R_0^{1/2}))^{-1}\}$, with a probability of at least $1-\delta$ on $(s, h_0) \sim \mu^m \otimes \rho_0$*

$$m\Psi\big(\mathcal{L}_s(h_K), \mathcal{L}_\mathcal{Z}(h_K)\big) \leq \frac{R_3}{\sigma_w^2\lambda_{\min}}(J_s(h_0) + R_2) - \sum_{k=0}^{K-1}\text{tr}\log\left(\text{Id} - \eta_k\nabla^2\mathcal{C}_s(h_k)\right) + \log\frac{2\xi}{\delta} + \delta,$$

*where $\xi = \int_{\mathcal{Z}^m \times \mathcal{H}} e^{m\Psi(\mathcal{L}_s(h), \mathcal{L}_\mathcal{Z}(h))}\mathrm{d}\mu^m(s)\mathrm{d}\rho_0(h)$.*

**Proof** Let $C$ be the constant given in Lemma 20 and suppose $n$ is sufficiently large so that its conclusion is satisfied. We construct the set $A_s$ as the set containing all points visited by gradient flows starting in $A(C, \delta, n)$. By construction, the entire trajectory $\{h_k\}_{k=0}^K$ lies in this set with probability at lest $1 - \delta$. To show that $\mathcal{C}_s$ is smooth on this set, we note that for any $h, h' \in$

$A_s(C, \delta, n)$,

$$\|\nabla \mathcal{C}_s(h) - \nabla \mathcal{C}_s(h')\|$$

$$\leq \frac{1}{m} \sum_{i=1}^{m} \left( |F_h(x_i) - y_i| \|\nabla F_h(x_i) - \nabla F_{h'}(x_i)\| + \|\nabla_h F_{h'}(x_i)\| |F_h(x_i) - F_{h'}(x_i)| \right)$$

$$\leq \frac{1}{m} \sum_{i=1}^{m} \left( |F_h(x_i) - y_i| \|\nabla F_h(x_i) - \nabla F_{h'}(x_i)\| + \|\nabla F_{h'}(x_i)\| \|\nabla F_{\hat{h}}(x_i)\| \|h - h'\| \right),$$

for some $\hat{h} \in \{\tau h + (1 - \tau)h' : \tau \in [0, 1]\}$.

From Lemma 19, it follows that $\|\nabla F_{h'}(x_i)\|_{\mathrm{F}} \leq \sqrt{n}R$. In fact, this holds for all parameter values of distance $Cn^{-1/2}$ from $A(C, \delta, n)$. Since this is a convex set which both $h$ and $h'$ belong to, $\hat{h}$ must belong to this set also, and so $\|\nabla F_{\hat{h}}(x_i)\|_{\mathrm{F}} \leq \sqrt{n}R$. Additionally, we apply Lemma 19 as well as the Cauchy-Schwarz inequality to deduce,

$$\frac{1}{m} \sum_{i=1}^{m} |F_h(x_i) - y_i| \|\nabla F_h(x_i) - \nabla F_{h'}(x_i)\| \leq \sqrt{n}R\mathcal{C}_s(h)^{1/2}\|h - h'\| \leq \sqrt{n}RR_0^{1/2}\|h - h'\|.$$

From this, we deduce that

$$\|\nabla \mathcal{C}_s(h) - \nabla \mathcal{C}_s(\tilde{h})\| \leq (R_0^{1/2}R\sqrt{n} + R^2 n)\|h - h'\| \leq R(R_0^{1/2} + R)n\|h - h'\|,$$

which means that the optimisation objective is $M$-smooth on $A_s(C, \delta, n)$, with $M = R(R_0^{1/2}+R)n$.

Since, by hypothesis, we consider a schedule such that $\sup_k \eta_k \leq 1/(2M)$, Theorem 4 applies. Moreover, we have

$$\log \frac{\rho_0(h_0)}{\rho_0(h_k)} \leq \frac{n}{2\sigma_\omega^2} \left| \|h_k\|^2 - \|h_0\|^2 \right| \leq \frac{n}{2\sigma_\omega^2} \|h_k - h_0\|(\|h_k - h_0\| + 2\|h_0\|).$$

Thus, using Lemma 21,

$$\log \frac{\rho_0(h_0)}{\rho_0(h_k)} \leq \frac{R_3}{\sigma_\omega^2 \lambda_{\min}}(J_s(h_0) + R_2).$$

from which, the statement follows. ∎

As noted in Lee et al. (2020) and Yang and Littwin (2021), the NTK analysis of wide neural networks can be performed in settings where the hidden layers have different widths and on networks with different architectures. Therefore, we should expect a similar argument to that given above can be reproduced in these settings.

**Remark 22 (Discussion on $J$)** *As noted in the proof of Lemma 21, the object $J$ defined in* (13) *can be written in the following form:*

$$J_s(h)^2 = \frac{1}{m} \sum_{i=1}^{m} \left( \sum_{l=1}^{L} \langle \delta_h^l(x_i), f_h^l(x_i) \rangle \right)^2.$$

*Thus, from Lemma 19, we readily obtain the bound,* $|\langle \delta_{h_0}^l(x_i), f_{h_0}^l(x_i) \rangle| \leq R_1^2 \sqrt{n}$ *for all* $h_0 \in A(C, \delta, n)$ *and thus,*

$$J_s(h_0) \leq L R_1^2 \sqrt{n}, \quad \text{for all } h_0 \in A(C, \delta, n). \tag{18}$$

*However, by borrowing arguments used in the analysis of the limiting form of the NTK, one could be convinced that this estimate is pessimistic. We provide an outline for the reasoning here – a formal analysis would require a careful investigation of the neural network at initialisation and is beyond the scope of this work. In the analysis of the limiting form of the NTK, the approach taken typically has the vectors computed in the forward pass, such as $f_{h_0}^l$, and the backwards pass, such as $\delta_{h_0}^l$, approximated by a zero mean Gaussian distribution. A critical theoretical tool used in this literature is the gradient independence assumption, which requires that the transpose of any weight matrix is independent to the forward pass in the limit as $n \to \infty$ (see, for example, Yang (2020)). This property is used to show that $\sqrt{n}\delta_{h_0}^l$ converges weakly to a zero mean Gaussian with diagonal covariance (see Yang and Littwin (2021)). Under this assumption, it also follows that the covariance in the vector $\sqrt{n}\delta_{h_0}^l(x_i) \odot f_{h_0}^l(x_i)$ is diagonal. Thus, the inner product could be approximated as follows:*

$$|\langle \delta_{h_0}^l(x_i), f_{h_0}^l(x_i)\rangle|^2 = \frac{1}{n}\Big(\sum_{i=1}^n \sqrt{n}(\delta_{h_0}^l(x_i))_i(f_{h_0}^l(x_i))_i\Big)^2$$

$$\approx \frac{1}{n}\sum_{i=1}^n \Big(\sqrt{n}(\delta_{h_0}^l(x_i))_i(f_{h_0}^l(x_i))_i\Big)^2.$$

*Since the entries of $f_{h_0}^l(x_i)$ and $\sqrt{n}\delta_{h_0}^l(x_i)|f_{h_0}^l(x_i)$ are identically distributed, we should thus expect that the sum converges with $n$ increasing. However, proving this formally would require an analysis of the rate with which the covariance decays due to gradient independence.*

**Remark 23 (Discussion on the curvature term)** *In Jacot et al. (2020), the authors study the Hessian of wide neural networks, showing that the Hessian takes the decomposition $\frac{1}{n}\nabla\mathcal{C}_s(h) = I(h) + S(h)$ where $I$ is a positive semi-definite matrix that has the same non-zero eigenvalues as the neural tangent kernel and $S$ is a matrix with a trace given by*

$$\text{tr}(S(h)) = \frac{1}{m}\sum_{i=1}^m \Delta_h F_h(x_i)(F_h(x_i) - y_i).$$

*They analyse these matrices in the limit as $n \to \infty$, showing that for any $r > 2$, $\text{tr}(S^r)$ vanishes. They also show that the matrices become orthogonal and thus, for any $r \in \mathbb{N}$,*

$$\text{tr}((I(h_k) + S(h_k))^r) \to \text{tr}(I_\star^r) + \text{tr}(S_\star(h_k)^r), \quad \text{as } n \to \infty.$$

*With this, it follows from the power series of the matrix logarithm, that with $n \to \infty$ we obtain the limit,*

$$\text{tr}\log\Big(\text{Id} -\eta\nabla^2\mathcal{C}_s(h_k)\Big) \to \text{tr}\log\Big(\text{Id} -\bar{\eta}I_\star\Big) + \bar{\eta}\,\text{tr}(S_\star(h_k)) - \bar{\eta}^2\,\text{tr}(S_\star(h_k)^2).$$

*Using the fact that $\text{tr}(S_\star(h_k)^2)$ is positive and that $I_\star$ is positive semi-definite, we can control this further using Lemma 5:*

$$\text{tr}\log\Big(\text{Id} -\bar{\eta}I_\star\Big) + \bar{\eta}\,\text{tr}(S_\star(k)) - \bar{\eta}^2\,\text{tr}(S_\star(k)^2) \le \bar{\eta}\frac{3\,\text{tr}(\Theta)}{2m} + \bar{\eta}\,\text{tr}(S_\star(k)).$$

*They further analyse the matrix $S_\star$ in the setting where the gradient flow is used. They show that the trace decays to zero over the iterations and at initialisation, its distribution is captured by a product of joint Gaussian matrices.*

## Appendix D. The analysis for general iterative methods

The proof of Theorem 9 follows immediately from the proof technique of Theorem 4 as soon as $-\nabla \mathcal{C}_s$ is replaced by the vector field. Similarly, we obtain the following result for the continuous-time flow dynamics.

**Theorem 24** *Consider the dynamics $\partial_t h_t = V_s(h_t; t)$, with $V_s : \mathcal{H} \times \mathbb{R}^+ \to \mathcal{H}$ differentiable in $h$. Let $\Psi : \mathbb{R}^2 \to \mathbb{R}$ be a measurable function. For any $\delta \in (0, 1)$ and $T > 0$, with probability at least $1 - \delta$ on the random draw $(s, h_0) \sim \mu^m \otimes \rho_0$, we have*

$$\Psi(\mathcal{L}_s(h_T), \mathcal{L}_{\mathcal{Z}}(h_T)) \leq \frac{1}{m} \left( \log \frac{\rho_0(h_0)}{\rho_0(h_T)} - \int_0^T \nabla \cdot V_s(h_t; t) \mathrm{d}t + \log \frac{\xi}{\delta} \right) ,$$

*where $\nabla$ refers to derivatives with respect to $h$ and $\xi = \int_{\mathcal{Z}^m \times \mathcal{H}} e^{m\Psi(\mathcal{L}_s(h), \mathcal{L}_{\mathcal{Z}}(h))} \mathrm{d}\mu^m(s) \mathrm{d}\rho_0(h)$.*

### D.1. Mini-batches

We can consider here a version of gradient descent that only evaluates the training objective on a mini-batch $s_k \subset s$ at each iteration $k \in \mathbb{N}$, as already discussed in Section 6. Note that the choice of the mini-batch can be random and doesn't need to be known a priori. Our result will always apply on the specific realisation of the sequence of batches used for the training.

We consider an objective in the form

$$\mathcal{C}_s(h) = \frac{1}{m} \sum_{z \in s} \hat{\ell}(z, h) ,$$

for some surrogate loss function $\hat{\ell} : \mathcal{Z} \times \mathcal{H} \to \mathbb{R}$. For a batch $s_k \subset s$ of $m_k$ elements, we write

$$\mathcal{C}_{s_k}(h) = \frac{1}{m_k} \sum_{z \in s_k} \hat{\ell}(z, h) .$$

**Proposition 25** *For a sequence of batches $\{s_k\}$, consider the dynamics $h_{k+1} = h_k - \eta_k \mathcal{C}_{s_k}(h_k)$. We denote as $\widetilde{\mu}^m$ the law of $(s, \{s_k\})$, which takes into account the potential randomness in the choice of the batches. For each $k$, let $A_s$ be a Borel where $\hat{\ell}(z, \cdot)$ is twice differentiable and $M$-smooth for every $z$ in $s$, with $\max_k \eta_k \leq 1/(2M)$. Let $\Psi : \mathbb{R}^2 \to \mathbb{R}$ be a measurable function. Fix $K \in \mathbb{N}$ and let $\delta \in (0, 1)$, such that the trajectory $\{h_k\}_{k=0}^{K-1}$ lies in $A_s$, with probability at least $1 - \delta/2$ on the randomness of the training dataset $s$, the initialisation $h_0$, and the choice of the batches. With probability at least $1 - \delta$ on the same randomness, we have*

$$\Psi(\mathcal{L}_s(h_K), \mathcal{L}_{\mathcal{Z}}(h_K)) \leq \frac{1}{m} \left( \log \frac{\rho_0(h_0)}{\rho_0(h_K)} - \sum_{k=0}^{K-1} \mathrm{tr} \log \left( \mathrm{Id} + \eta_k \nabla^2 \mathcal{C}_{s_k}(h_k) \right) + \log \frac{2\xi}{\delta} + \delta \right) ,$$

*with $\xi = \int_{\mathcal{Z} \times \mathcal{H}} e^{m\Psi(\mathcal{L}_{\bar{s}}(h), \mathcal{L}_{\mathcal{Z}}(h))} \mathrm{d}\mu^m(\bar{s}) \mathrm{d}\rho_0(h)$.*

### D.2. Momentum

We can consider the use of auxiliary variables: instead of having just $h_k$, we take the pair of processes $(h_k, v_k)$. If the updates of these processes are of the form

$$\begin{pmatrix} h_{k+1} \\ v_{k+1} \end{pmatrix} = \begin{pmatrix} h_k \\ v_k \end{pmatrix} + V_s(h_k, v_k; k),$$

for some iteration-dependent vector field $V_s$, then the usual analysis applies immediately. For example, let us consider the momentum scheme

$$h_{k+1} = h_k + v_{k+1}, \qquad v_{k+1} = \mu_k v_k - \eta_k \nabla \mathcal{C}_s(h_k),$$

where $\mu_k \in [0, 1]$ is the momentum schedule.

This corresponds to the vector field

$$V_s(h, v; k) = \begin{pmatrix} \mu_k v - \eta_k \nabla \mathcal{C}_s(h) \\ (\mu_k - 1)v - \eta_k \nabla \mathcal{C}_s(h) \end{pmatrix},$$

whose Jacobian reads

$$\nabla V_s(h, v; k) = \begin{pmatrix} -\eta_k \nabla^2 \mathcal{C}_s(h) & \mu_k \, \mathrm{Id} \\ -\eta_k \nabla^2 \mathcal{C}_s(h) & (\mu_k - 1) \, \mathrm{Id} \end{pmatrix}.$$

We can easily compute

$$\det(\mathrm{Id} + \nabla V_s(h, v; k)) = \det \left( \mu_k(\mathrm{Id} - \eta_k \nabla^2 \mathcal{C}_s(h)) + \mu_k \eta_k \nabla^2 \mathcal{C}_s(h) \right) = \det(\mu_k \, \mathrm{Id}) = \mu_k{}^d,$$

and so

$$-\sum_{k=0}^{K-1} \mathrm{tr} \log \left( \mathrm{Id} + \nabla V_s(h_k, v_k; k) \right) = -\sum_{k=0}^{K-1} \log \det \left( \mathrm{Id} + \nabla V_s(h_k, v_k; k) \right) = d \sum_{k=0}^{K-1} \log \frac{1}{\mu_k}.$$

If we consider the schedule $\mu_k = 1 - \alpha(k+1)^{-1}$, for some fixed $\alpha < 1$, then we obtain that this sum scales with $\alpha d \log K$, and this is made dimension independent by choosing $\alpha \sim 1/d$. Curiously, when $\mu_k \equiv 1$, the sum vanishes. As a final remark, note that one must initialise the pair $(h_0, v_0)$ by drawing it from a fixed distribution $\rho_0$, that we assume to have full support on $\mathcal{H} \times \mathbb{R}^d$. This excludes the case of a deterministic initial value for the velocity.

## Appendix E. Discretised damped Hamiltonian dynamics

Here, we consider a Hamiltonian approach, and hence we introduce $d$ additional variables $v$, representing the velocities (momenta) of the parameters $h$. The idea is to exploit the fact that the joint density of the pair $(h, v) \in \mathcal{H}^2$ is conserved under the Hamiltonian flow, a property that is preserved for discrete time-steps by suitable symplectic integrators (Hairer et al., 2006). In order to solve an optimisation problem, we can alternate conservative Hamiltonian steps with dissipative ones, which only involve $v$ and entail an exactly computable change in density.

Let us make things more concrete. For the rest of this section, we denote as $\rho_k$ the joint density of $(h_k, v_k)$. We consider an increasing differentiable map $\psi : \mathbb{R} \to \mathbb{R}$, such that $\psi(0) = 0$, and

we fix $\eta > 0$. We denote as $\Psi_\eta(v)$ the value at $t = \eta$ of the solution of $\partial_t \tilde{v}_t = -\psi(\tilde{v}_t)$ satisfying $\tilde{v}_0 = v_k$, where with a slight abuse of notation we are here implying that $\psi$ is acting component-wise on $\tilde{v}_t \in \mathbb{R}^d$. From $(h_k, v_k)$, to evaluate $(h_{k+1}, v_{k+1})$ we first proceed with a dissipative step:

$$h_{k+1/2} = h_k \, ; \qquad\qquad v_{k+1/2} = \Psi_\eta(v_k) \, .$$

Since this step involves the exact solution of a continuous-time gradient descent evolution, we can appeal to the usual continuity arguments to show that

$$\log \frac{\rho_{k+1/2}(h_{k+1/2}, v_{k+1/2})}{\rho_k(h_k, v_k)} = \sum_{i=1}^d \log \frac{\psi(v_k^i)}{\psi(v_{k+1/2}^i)} \, , \tag{19}$$

with $v^i$ denoting the $i$-th component of $v$. Indeed, we see that for each component of $\tilde{v}_t$

$$\psi'(\tilde{v}_t^i) = \frac{\partial_t(\psi(\tilde{v}_t^i))}{\partial_t \tilde{v}_t^i} = -\frac{\partial_t(\psi(\tilde{v}_t^i))}{\psi(\tilde{v}_t^i)} = -\partial_t(\log \psi(\tilde{v}_t^i)) \, ,$$

and so

$$\int_0^\eta \psi'(\tilde{v}_t^i) \mathrm{d}t = \log \frac{\psi(\tilde{v}_0^i)}{\psi(\tilde{v}_\eta^i)} = \log \frac{\psi(v_k^i)}{\psi(v_{k+1/2}^i)} \, ,$$

from which (19) follows.

After this dissipative step, we apply a symplectic Hamiltonian integrator, such as

$$h_{k+1} = h_{k+1/2} + \eta v_{k+1} \, ; \qquad\qquad v_{k+1} = v_{k+1/2} - \eta \nabla_h \mathcal{C}_s(h_k) \, .$$

This step conserves the density:

$$\rho_{k+1}(h_{k+1}, v_{k+1}) = \rho_{k+1/2}(h_{k+1/2}, v_{k+1/2}) \, .$$

Indeed, we are applying to $(h_k, v_k)$ the tranformation

$$\begin{pmatrix} h \\ v \end{pmatrix} \mapsto \begin{pmatrix} h + \eta v - \eta^2 \nabla \mathcal{C}_s(h) \\ v - \eta \nabla \mathcal{C}_s(h) \end{pmatrix} \, ,$$

whose Jacobian $\begin{pmatrix} 1 - \eta^2 \Delta \mathcal{C}_s(h) & \eta \\ -\eta \Delta \mathcal{C}_s(h) & 1 \end{pmatrix}$ has determinant 1 for all $h$ and $v$.

Following the above dynamics for $K$ steps and applying the usual Markov argument we obtain the following result.

**Proposition 26** *Consider the damped Hamiltonian dynamics described above, with $\mathcal{C}_s$ twice differentiable on the whole $\mathcal{H}$. Let $\Psi : \mathbb{R}^2 \to \mathbb{R}$ be a measurable function. Fix $K \in \mathbb{N}$ and let $\delta \in (0, 1)$. With probability at least $1 - \delta$ on the random draw $(s, h_0, v_0) \sim \mu^m \otimes \rho_0$, we have*

$$\Psi(\mathcal{L}_{\mathcal{Z}}(h_K), \mathcal{L}_s(h_K)) \le \frac{1}{m} \left( \log \frac{\xi}{\delta} + \log \frac{\rho_0(h_0, v_0)}{\rho_0(h_K, v_K)} + \sum_{k=0}^{K-1} \sum_{i=1}^d \log \frac{\psi(v_k^i)}{\psi(v_{k+1/2}^i)} \right) \, , \tag{20}$$

*with $\xi = \int_{\mathcal{Z}^m \times \mathcal{H}^2} e^{m\Psi(\mathcal{L}_{\bar{s}}(h), \mathcal{L}_{\mathcal{Z}}(h))} \mathrm{d}\mu^m(\bar{s}) \mathrm{d}\rho_0(h, v)$.*

As a concrete example, we can choose $\psi(v) = \varepsilon|v|^p v$, for $p \geq 0$ and $\varepsilon > 0$, where $|\cdot|$ denotes the absolute value computed component-wise. The case $p = 0$ corresponds to the standard conformal damped Hamiltonian dynamics (França et al., 2020), and yields to

$$h_{k+1} = h_k + \eta v_{k+1}\,; \qquad\qquad v_{k+1} = e^{-\varepsilon\eta}v_k - \eta\nabla_h\mathcal{C}_s(h_k)\,,$$

with a density that increases exponentially as

$$\log \frac{\rho_{k+1}(h_{k+1}, v_{k+1})}{\rho_k(h_k, v_k)} = d\varepsilon\eta\,.$$

Note that this last term goes linearly with the dimension of the hypothesis space, a behaviour that is likely to bring poor bounds in over-parameterised settings. To avoid this, one can choose $p > 0$ and get

$$h_{k+1} = h_k + \eta v_{k+1}\,; \qquad\qquad v_{k+1} = \frac{v_k}{(1 + p\varepsilon\eta|v_k|^p)^{1/p}} - \eta\nabla_h\mathcal{C}_s(h_k)\,,$$

and

$$\log \frac{\rho_{k+1}(h_{k+1}, v_{k+1})}{\log \rho_k(h_k, v_k)} = \left(1 + \frac{1}{p}\right)\sum_{i=1}^{d}\log\left(1 + p\varepsilon\eta|v_k^i|^p\right)\,.$$

With this choice, if the components of $v$ are smaller than 1 (*e.g.*, when they are sampled from a Gaussian with small variance) the last term in the RHS of (20) will likely have a better behaviour with $d$. However, this improvement might come at the price of a larger $\log\frac{\rho_0(h_0, v_0)}{\rho_0(h_K, v_K)}$, due to the fact that less dissipative dynamics allow the model to explore a wider region of the hypothesis space, potentially ending up with a final state $(h_K, v_K)$ with a $\rho_0$ density much lower than $\rho_0(h_0, v_0)$. It is unclear so far what is the optimal choice of $\psi$ that can lead to tightest bounds. We leave the investigation of this open problem as future work.

## Appendix F. A comment on the Laplacian's integral

An interesting feature of the bound of Theorem 1 is the integral of the Laplacian of the optimisation objective along the training path. Under the gradient flow dynamics, $\Delta\mathcal{C}_s(h_t)$ is keeping track of how the probability density is locally varying. Indeed, from a PAC-Bayesian perspective, for the bound to be small we need to end up in some point where the posterior density is not too high compared to the initial one. If we follow a trajectory characterised by a large Laplacian, we see a sharp increase of the density. Intuitively, we can picture this situation as if we were attracting the nearby paths and bringing further *probability mass* around us. In such a case, the final $\rho_T$ is likely to be much larger than the initial $\rho_0$ and lead to a loose bound. Note that we can rewrite

$$\int_0^T \Delta\mathcal{C}_s(h_t)\mathrm{d}t = \log\frac{\|\nabla\mathcal{C}_s(h_0)\|}{\|\nabla\mathcal{C}_s(h_T)\|} - \int_{h_{[0:T]}} \nabla\cdot\tau(h)\,\|\delta h\|\,, \tag{21}$$

(see below for a derivation). Here $\tau(h) = -\frac{\nabla\mathcal{C}_s(h)}{\|\nabla\mathcal{C}_s(h)\|}$ is the unit tangent vector to the trajectory in $h$, and $\int_{h_{[0:T]}}\ldots\|\delta h\|$ denotes the line integral along the path $h_{[0:T]}$. The last term has a clear meaning, as it quantifies how much $h_{[0:T]}$ is attracting the nearby trajectories.

To obtain (21), first notice that $\partial_t \log \|\nabla \mathcal{C}_s(h_t)\| = -\frac{\nabla \mathcal{C}_s(h_t)}{\|\nabla \mathcal{C}_s(h_t)\|} \cdot \nabla \|\nabla \mathcal{C}_s(h_t)\|$. Since, for all $h$,

$$\Delta \mathcal{C}_s(h) = \nabla \cdot \nabla \mathcal{C}_s(h) = \nabla \cdot \left( \frac{\nabla \mathcal{C}_s(h)}{\|\nabla \mathcal{C}_s(h)\|} \right) \|\nabla \mathcal{C}_s(h)\| + \frac{\nabla \mathcal{C}_s(h)}{\|\nabla \mathcal{C}_s(h)\|} \cdot \nabla \|\nabla \mathcal{C}_s(h)\| \,,$$

we get

$$\Delta \mathcal{C}_s(h_t) = \nabla \cdot \left( \frac{\nabla \mathcal{C}_s(h_t)}{\|\nabla \mathcal{C}_s(h_t)\|} \right) \|\nabla \mathcal{C}_s(h_t)\| - \partial_t \log \|\nabla \mathcal{C}_s(h_t)\| \,.$$

We conclude that

$$\int_0^T \Delta \mathcal{C}_s(h_t) \mathrm{d}t = \log \frac{\|\nabla \mathcal{C}_s(h_0)\|}{\|\nabla \mathcal{C}_s(h_T)\|} + \int_0^T \nabla \cdot \left( \frac{\nabla \mathcal{C}_s(h_t)}{\|\nabla \mathcal{C}_s(h_t)\|} \right) \|\nabla \mathcal{C}_s(h_t)\| \mathrm{d}t \,.$$

The integral in the RHS is a line-integral along the path $h_{[0,T]}$, as $\|\nabla \mathcal{C}_s(h)\|$ is the norm of the flow's "velocity" in $h$. Moreover, $\tau(h) = -\frac{\nabla \mathcal{C}_s(h)}{\|\nabla \mathcal{C}_s(h)\|}$ is the unit tangent vector to the gradient flow in $h$. We can thus write

$$\int_0^T \Delta \mathcal{C}_s(h_t) \mathrm{d}t = \log \frac{\|\nabla \mathcal{C}_s(h_0)\|}{\|\nabla \mathcal{C}_s(h_T)\|} - \int_{h_{[0:T]}} \nabla \cdot \tau(h) \|\delta h\| \,.$$

## Appendix G. Implicit regularisation in first-order methods

In this section, we consider some modifications of gradient descent that are known to lead to better generalisation and we relate this improvement to the bound given in Theorem 4.

The role of noise in both optimisation and generalisation is a topic that has received considerable attention recently. The Langevin dynamics, as well as its discrete-time approximations, have proved to be a popular model for investigating this (Raginsky et al., 2017; Mou et al., 2018; Erdogdu et al., 2018; Farghly and Rebeschini, 2021). The Langevin diffusion is most often described by

$$\mathrm{d}h_t = -\nabla \mathcal{C}_s(h_t) \mathrm{d}t + \sigma \mathrm{d}B_t \,, \tag{22}$$

where $B_t$ is a Brownian motion. Its generalisation properties have been studied using uniform stability and information-theoretic bounds (see Sections 7.2 and 7.3). The associated Fokker-Planck equation yields that the evolution of the marginal density is identical to that of the deterministic flow

$$\partial_t \hat{h}_t = -\nabla \left( \mathcal{C}_s(\hat{h}_t) + \frac{\sigma^2}{2} \log \hat{\rho}_t(\hat{h}_t) \right) \,; \quad \hat{h}_0 \sim \rho_0 \,, \tag{23}$$

where we denote as $\hat{\rho}_t$ the density of $\hat{h}_t$ (Pavliotis, 2014). This idea was recently exploited in Song et al. (2021) in the context of generative modelling. We note that Theorem 1, with the choice of $\Psi(u,v) = \frac{2}{m\sigma^2}(v-u)$ (with $\sigma > 0$) and with a bounded loss $\ell \subseteq [0,1]$, leads to the bound

$$\mathcal{L}_{\mathcal{Z}}(h_T) \leq \mathcal{L}_s(h_T) + \frac{1}{4m\sigma^2} + \frac{\sigma^2}{2} \left( \log \frac{\rho_T(h_T)}{\rho_0(h_T)} + \log \frac{1}{\delta} \right) \,.$$

Using this as optimisation objective is equivalent to following (23) with $\mathcal{C}_s = \mathcal{L}_s - \frac{\sigma^2}{2} \log \rho_0$.

On another note, applying uniformly distributed label noise during training can improve generalisation (Blanc et al., 2020; Damian et al., 2021). Damian et al. (2021) established that in certain

settings this is characterised by the implicit regularisation term $-\frac{\sigma^2}{2B}\operatorname{tr}\log(1-\frac{\eta}{2}\nabla^2 \mathcal{C}_s)$ that is added to the optimisation objective. Here, $\sigma$ is the scale of the label noise and $B$ the batch size. Up to a scaling factor, this is precisely the term that is summed over in the bound of Theorem 4.

However, we note that these algorithms are not optimising our PAC-Bayes bounds. Indeed, if one were to try to use the bound as an optimisation objective, our generalisation guarantee would change (as it would involve the Hessian of this new objective). We defer to future work for deeper analysis of how our bounds relate to the generalisation properties of these algorithm.

