# OpenReview forum: "Generalisation under gradient descent via deterministic PAC-Bayes"
_algorithmiclearningtheory.org/ALT/2025/Conference — ALT 2025_

### Official Review · Reviewer_oEqj · 2024-10-28

**Rating:** 6
**Confidence:** 4

**Review:**

This paper develops generalization bounds for gradient descent methods and continuous gradient flows. Unlike the standard PAC-Bayesian bounds, the derived bounds are stated with high probability with respect to both the training dataset and the randomness of the initial model. The basic idea is that the density function of the trajectory can be expressed in terms of the Hessian matrix or Laplacian operator. In this way, there is no need to introduce a derandomisation step. The paper gives generalization bounds of order $O(1/m)$ if small empirical loss can be achieved, where $m$ is the sample size. Applications to specific models are then given, including random feature model and wide neural networks.

**Strength**
- The developed generalization bounds show advantages over existing PAC-Bayesian bounds as they are stated with high probability with the dataset and the initialization. Also, there is no need to introduce a derandomisation step.
- The paper gives comparisons with three popular approaches: PAC-Bayesian approach, stability approach and information-theoretical approach. As compared to the stability approach, the paper does not require a Lipschitzness assumption and can imply faster rates. As compared to the information-theoretical approach which implies bounds in expectation, the bounds in this paper depend directly on the optimisation trajectory.
- The developed generalization bounds are general in the sense that it considers a general $\Psi$ and $\nabla C_s$. Extensions to stochastic gradient descent and momentum dynamics are also given.

**Weakness**
- A downside of the bounds in the paper is that the derived generalization bounds look complicated. For example, in Theorem 4, the bounds involve $tr log (I-\eta\nabla^2C_s)$. It may be difficult to give a tight estimate for this term. While the paper shows that this can be bounded by the trace norm of the Hessian matrix in Lemma 5, this leads to a linear dependency on $d$. Therefore, the derived bounds may not be appealing for high-dimensional settings.
- For the applications to neural networks, the derived generalization bounds depend on $n\sum_k\eta_k \land \lambda_{min}^{-1}$. Note that $n$ can be very large, and then the term $n\sum_k\eta_k$ would be very large. Furthermore, the smallest eigenvalue is often very small, and then the term $\lambda_{min}^{-1}$ is also very large. Therefore, the generalization bound in Proposition 8 may not be quite effective due to $n\sum_k\eta_k \land \lambda_{min}^{-1}$. It is also interesting to state clearly how large $n_{\min}^\delta$ should be in Proposition 8.
- Several steps are not quite clear to me. For example, it would be beneficial to give some explanations on the equation $\partial_t\rho_t(h)=\nabla\cdot(\rho_t(h)\nabla C_s(h))$. Also, the notation $G_{\eta}$, $\widetilde{G}_k$ in the proof of Theorem 4 are not clear to me. It is also note clear to me how the equality $\tilde{G}_k(h_k)=h_{k+1}$ for $_k\in A_s$.

**Typos**:
- Theorem 4: "smoothwhere" should be "smooth, where"
- Section 6, the part with SGD: "Id+\eta_k" should be $Id-\eta_k$
- Proof of Corollary 3: "," should be in the displayed equation

**Paper Award:**

No

---

> ### Author Response · Authors · 2024-11-25
>
> We thank the reviewer for the positive feedback and the careful and detailed review. We address below the concerns raised.
>
>  - We agree with the reviewer that the term $\mathrm{tr}\log ( \mathrm{Id} - \eta_k \nabla^2 \mathcal{C}_s(h_k))$ can be hard to upper bound in general. However, it is important to emphasise that this term can, in principle, be computed exactly (albeit with a potentially high computational cost for over-parameterised models) since it only requires knowledge of the second derivative of the training objective at the points $h_k$ visited along the trajectory. The upper bounds that we provide might indeed be pretty loose in general. For instance, if the learning rates tend to zero, the term will converge to minus the Laplacian of the training objective (plus a correction term of order $\eta^2$), as shown in Lemma 5. As the Laplacian can be much smaller than the trace norm, this last upper bound in general can be quite loose. We agree that deriving simpler and tighter bounds would further enhance the work, and we plan to explore this direction in future research.
>
> - On the subject of the NTK example, the object that R3 is referring to as $n \sum_k \eta_k \wedge \lambda_\min^{-1}$ is in fact the object $A_K(h_0) = ( \sum_k \eta_k \wedge \lambda_\min^{-1})(J_s(h_0)+1)$ which appears in the upper bound divided by $m$. We provide a bound to show that $J_s(h_0)$ scales no faster than $\sqrt{n}$ (as oppose to $O(n)$) but we also stress here that this bound is naive and in Remark 22 we provide a heuristic argument showing that under the gradient independence assumption (which is commonly used in the NTK literature) we can improve this bound to scale as $O(1)$. As a result, this term would contribute $( \sum_k \eta_k \wedge \lambda_\min^{-1})/m$ to the bound. However, proving this tighter bound formally would require significant work on the convergence of wide NNs which is far beyond the scope of this work. The reviewer is also correct in highlighting the potential issues with using the minimum eigenvalue of the kernel — this does indeed have the potential scale poorly with $n$. However, the minimum eigenvalue is a quantity of significant importance in the NTK analysis, as it provides worst-case bounds on the convergence of the associated gradient flow. We agree that further analysis and further assumptions could lead to improvements on this front, however, such work would require significant work on the NTK which we believe is far beyond the scope of this work. Furthermore, we emphasise that the minimum eigenvalue appears in a variety of generalisation bounds for the NTK setting, and since the purpose of section 5.2 is to explore how our methodology (which is quite different to existing approaches) interacts with the machinery developed in the NTK literature, we believe that the current result is sufficient. Overall, the current proof, presented in section C.2, relies heavily on existing results in the NTK literature and while we agree with the reviewer that improvements can be made, we believe that many of the criticisms stem more fundamentally from the approach of the NTK.
>
>  - We agree with the reviewer that the continuity equation may require additional explanation for readers less familiar with the concept, especially given its central role in our results. This equation is a fundamental result in the field of PDEs. The most intuitive interpretation probably comes from fluid dynamics, where it encodes the principle that the change in the amount of fluid within a fixed volume equals the difference between the inflow and the outflow. It can also be viewed as a noiseless version of the Fokker-Planck equation, where diffusion is absent. We will include in the revised version of our paper some clarifying remarks to give some more intuition, and add more references.
>
>  - We will try to simplify the proof of Thm 4, or at least to justify more clearly the quantities and notation introduced there. The introduction of $A_s$ is in order to have the bijectivity of the steps, which allows the use of the change of variable formula. $G_\eta$ is simply the map that maps a point to the next one under the GD algorithm. In general this is not bijective, but its restriction to $A_s$ is. If one starts from $A_s^0$, the whole trajectory will lie in $A_s$, and hence can be seen as a composition of bijections. These bijections are then denoted with the tilde notations. Regarding the statement $\tilde{G}_k(h_k) = h_{k+1}$, we thank the reviewer for pointing it out, as there is a slight imprecision there. It does not hold for any $h_k\in A_s$, but just for any $h_k\in \mathrm{supp}\tilde\rho_k^s$, as $\tilde G_k$ is only defined there. This however does not impact the rest of the proof, as if $h_0\in A_s^0$, then $h_k\in\mathrm{supp}\tilde\rho_k^s$, by definition of $\tilde\rho_k^s$.
>
>  - We thank the reviewer for pointing out a few typos, which we will fix for the revised version of the paper.

---

### Official Review · Reviewer_ffKT · 2024-11-09
**Review for paper "Generalisation under gradient descent via deterministic PAC-Bayes"**

**Rating:** 7
**Confidence:** 4

**Review:**

This paper studies generalization bounds for learning models that minimize some loss function (such as the empirical loss) that depends on some random training data. The authors propose a deterministic PAC-Bayes view where the randomness only comes from the training data and from the distribution of the initial point to the algorithm, but the training algorithm is deterministic such as gradient descent or gradient flow. The authors prove an upper bound to the generalization error that holds with high probability. Notably, the upper bound is a function of the iterates of the training algorithm, so in principle it can be computed in practice along the algorithm execution. The authors provide some examples where their result can give a tighter upper bound in some regime compared to existing bounds. The authors also extend their results to show upper bound guarantees for other iterative algorithms, including stochastic gradient descent, momentum methods, and damped Hamiltonian methods.

To prove the results, the authors use the continuity equation for the gradient flow (or the change of variable formula for the algorithms) to track the evolution of the log density of the distributions of the iterates along the algorithm. The authors then combine this with standard concentration argument using Markov inequality to derive the high-probability bound for the generalization error. The use of the continuity equation is natural given the dynamics, and it is interesting to see how to use it to bound the generalization error. The technique of tracking the change in distributions is very flexible and can be extended to any iterative algorithms, as the authors demonstrated.

This paper is well-written and provides a good overview of the problem, techniques and results, and comparison with related literature. This paper makes a concrete contribution to the theory of generalization error by providing a new upper bound which is instance-specific and seems to be better in some settings.

In Thm 4: Is s’ = s?

**Paper Award:**

No

---

> ### Author Response · Authors · 2024-11-25
>
> We thank the reviewer for their very positive feedback, and for appreciating the simplicity and generality of our framework. As for their concern about $s'$ in Thm 4, we agree that this is perhaps a confusing notation. The idea was to state that the initial assumption (the one involving $s'$) is valid for any dataset, which does not have to be the specific $s$ used for the training, which is drawn randomly. However, we will try to rephrase the statement in a less confusing and simpler way.

---

### Official Review · Reviewer_K9iN · 2024-11-09
**Interesting derandomization on general PAC-Bayes bounds**

**Rating:** 6
**Confidence:** 3

**Review:**

This paper proposes a new PAC-Bayesian generalization bound for models trained with gradient flow or gradient descent. Compared with previous de-randomized PAC-Bayesian bounds assuming specific model structures, this work’s analysis applies to a more general family of models. The authors prove a continuous-time bound first, then extend it to discrete time gradient descent. They also computed the explicit form of their bound for random feature model and NTK. Detailed comparisons with previous generalization bounds are provided.

Strengths:
- Obtaining a general PAC-Bayes bound for deterministic algorithms such as gradient descent is always an exciting topic.
- The paper is well written, with detailed comparison with previous literature.

Comments & Questions:
- It seems unclear how fast the PAC-Bayes bound converges. A general unconditional sample rate of $\tilde{O}(\frac{1}{m})$ should be intractable. I wonder when does Theorem 4 achieve this rate, and when does it imply only the slower rate of $\tilde{O}(\frac{1}{\sqrt{m}})$? It is argued that the faster rate is guaranteed assuming $\mathcal{L}_s\sim 1/m$, but the authors should justify why such an assumption is reasonable.
- How does the result compare with previous deterministic PAC-Bayes bounds such as [1]?
- In Section 7.1, the third line below eq (6), $R$ should be $\Gamma$.

References.
[1]. Luo, Xuanyuan, Bei Luo, and Jian Li. "Generalization bounds for gradient methods via discrete and continuous prior." Advances in Neural Information Processing Systems 35 (2022): 10600-10614.

**Paper Award:**

No

---

> ### Author Response · Authors · 2024-11-25
>
> We thank the reviewer for the positive review and for appreciating the generality of the framework we propose. We respond to their questions and comments below.
>
>  - We agree with the reviewer that, in the general agnostic setting, rates no better than $O(1/\sqrt m)$ can be expected. However, bounds adaptive to the variance can achieve faster rates. When $\ell$ is bounded in $[0,1]$, the variance is smaller than the population loss, which can be arbitrarily close to $0$ in the realisable setting. This is captured by the small-kl bound, where if the empirical loss converges to $0$ faster than $1/m$, the population loss as well is shown to converge at rate $O(1/m)$ (see for instance the discussion after Thm 1 in [1], or in Section 2 of [2]). Literature on optimisation gives conditions under which these fast rates of convergence of the empirical loss are achievable (e.g.,  Bernstein condition). We will add to the revised version of the paper a more detailed discussion on these points, stressing more that in general only slower rates can be achieved.
>
>  - We thank the reviewer for pointing out the work of Luo, Bei, and Li, which we were not previously aware of. While we agree that the paper is relevant to our work and will include it in our revised PAC-Bayes literature section, their framework and ours differ significantly, which might prevent a direct comparison of the results. Specifically, their analysis focuses on discretised versions of GD and SGD. Their bounds are stated in terms of the gradient discretisation error, and become vacuous as this error approaches zero (standard GD/SGD). Moreover, their bounds require the prior distribution to depend on a subset of the training dataset, as they include a term comparing the gradient computed on the subset associated with the prior with the gradient computed on the remainder of the dataset. The empirical loss appearing in their bound is evaluated only on the data points not used for the prior. We believe that our results address more conventional versions of GD and SGD, albeit with the trade-off of requiring a smoothness assumption not needed in their work.
>
>  - We thank the reviewer for pointing out the typo, which we will fix for the revised version of our paper.
>
>
> [1] Tolstikhin and Seldin, "PAC-Bayes-Empirical-Bernstein Inequality", 2013.
> [2] Mhammedi et al, "PAC-Bayes Un-Expected Bernstein Inequality", 2019.

---

### Author Rebuttal · Authors · 2024-11-25

We thank the reviewers for their time and constructive feedback on our submission, which we will incorporate to improve our paper. We respond to each reviewer separately below.

---

### Meta-Review · Area_Chair_gsR1 · 2024-12-12

**Recommendation:** Accept
**Confidence:** 4

**Metareview:**

This paper presents high-probability PAC-Bayes generalization bounds for models learned using either continuous-time gradient flows or discrete-time gradient descent. The analysis, using the disintegrated PAC-Bayes machinery, does not require any randomization in the algorithm apart from the random initialization. All the reviewers have acknowledged the clarity of the presentation, the novelty and strength of the results, and the originality of the proofs. This work is a solid contribution to the literature on generalization analysis of learning algorithms that takes into account the interaction between the algorithm, the loss function, and the data distribution.

**Paper Award:**

No